# How Does Topology Bias Distort Message Passing in Graph Recommender? A Dirichlet Energy Perspective

Yanbiao Ji[1]    Yue Ding[1]*    Dan Luo[2]    Chang Liu[1]
Yuxiang Lu[1]    Xin Xin[3]    Hongtao Lu[1]

[1]Shanghai Jiao Tong University    [2]Lehigh University    [3]Shandong University

{jiyanbiao, dingyue, isonomialiu, luyuxiang_2018, htlu}@sjtu.edu.cn
danluo.ir@gmail.com    xinxin@sdu.edu.cn

## Abstract

Graph-based recommender systems have achieved remarkable effectiveness by modeling high-order interactions between users and items. However, such approaches are significantly undermined by popularity bias, which distorts the interaction graph's structure—referred to as *topology bias*. This leads to overrepresentation of popular items, thereby reinforcing biases and fairness issues through the user-system feedback loop. Despite attempts to study this effect, most prior work focuses on the embedding or gradient level bias, overlooking how topology bias fundamentally distorts the message passing process itself. We bridge this gap by providing an empirical and theoretical analysis from a Dirichlet energy perspective, revealing that graph message passing inherently amplifies topology bias and consistently benefits highly connected nodes. To address these limitations, we propose **Test-time Simplicial Propagation** (TSP), which extends message passing to higher-order simplicial complexes. By incorporating richer structures beyond pairwise connections, TSP mitigates harmful topology bias and substantially improves the representation and recommendation of long-tail items during inference. Extensive experiments across five real-world datasets demonstrate the superiority of our approach in mitigating topology bias and enhancing recommendation quality. The implementation code is available at `https://github.com/sotaagi/TSP`.

## 1 Introduction

Recommender systems (RS) are fundamental components of modern online platforms, playing a crucial role in connecting users with relevant items [16, 64, 8]. Recently, graph-based methods, such as LightGCN [21], have gained prominence in RS, as they effectively capture collaborative filtering signals by modeling high-order connectivity between users and items [34, 66, 63, 72]. Despite their effectiveness, graph-based methods suffer from popularity bias, where popular items are recommended disproportionately more often than less popular items [1, 3]. This notorious effect not only undermines the accuracy and fairness of recommendation [74, 35, 55], but also exacerbates the Matthew Effect and the filter bubble [38, 45] through the user-system feedback loop.

In graph-based recommender systems, popularity bias mainly distorts the topology of the interaction graph, producing a highly biased degree distribution as shown in Figure 1(a), which we refer to as *topology bias*. While research has shown that the topology bias gets further amplified [56, 76] in graph-based methods, the root cause of this effect remains rather unexplored. Although some recent studies have attempted to understand this issue, they exhibit significant limitations. Their

---

*Corresponding Author

analyses focus mainly on the embedding landscape [28, 26] or gradient bias [10, 71, 31] *after message passing*. The question of how topology bias distorts the message passing process remains unsolved.

To bridge this research gap, we investigate the message passing process in graph-based RS from the Dirichlet energy perspective. Our empirical and theoretical analysis reveals the following results: (1) The graph message passing mechanism is equivalent to minimizing the global graph Dirichlet energy. (2) Popular nodes with high degrees concentrate embedding norms. (3) The update of the norm of a node's embedding is upper bounded by local Dirichlet energy.

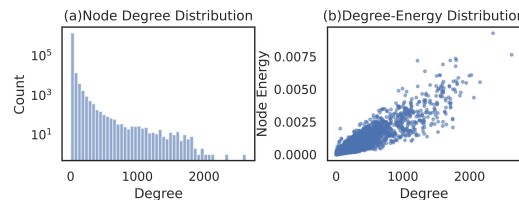

Figure 1: The degree and energy distribution of LightGCN embeddings on Gowalla dataset.

Based on the above studies, we propose **Test-time Simplicial Propagation** (TSP), which performs message passing over simplicial complexes with intra- and inter-simplex smoothing to avoid the dominance of certain nodes or substructures. By incorporating simplices instead of mere edges, our proposed TSP mitigates harmful topology bias during inference, improving the performance of tail nodes without requiring RS retraining.

We summarize our contributions below:

- We provide comprehensive theoretical analysis of how topology bias distorts message passing in graph-based recommender systems. We establish formal connections between graph message passing, Dirichlet energy optimization, and node embeddings, offering a mathematical framework to understand why popular nodes are systematically favored.
- We propose a principled approach to overcome the limitations of pairwise edge message passing by introducing TSP, which leverages higher-order simplicial complexes for information propagation, enabling more balanced representation learning across nodes of varying popularity.
- We develop TSP as a plug-and-play solution that operates solely during inference, making it compatible with existing recommender systems without requiring costly retraining.
- Through extensive experiments across five diverse real-world datasets, we demonstrate that TSP can effectively mitigate topology bias without impairing the overall recommendation quality. Our results show consistent performance gains in tail nodes as well as competitive or better results in overall recommendation compared to state-of-the-art baselines.

## 2 Preliminaries

### 2.1 Graph-based Recommendation

Let $\mathcal{U} = \{u_1, u_2, \ldots, u_{|\mathcal{U}|}\}$ denote the set of users and $\mathcal{I} = \{i_1, i_2, \ldots, i_{|\mathcal{I}|}\}$ denote the set of items. User-item interactions are represented as an implicit feedback matrix $\mathbf{R} \in \{0,1\}^{|\mathcal{U}| \times |\mathcal{I}|}$, where $\mathbf{R}_{ui} = 1$ if user $u$ interacted with item $i$, and 0 otherwise. These interactions can be naturally modeled as a graph $\mathcal{G} = (\mathcal{V}, \mathcal{E})$, specifically a user-item bipartite graph where the node set $\mathcal{V} = \mathcal{U} \cup \mathcal{I}$ and the edge set $\mathcal{E}$ contains edges $(u, i)$ if and only if $\mathbf{R}_{ui} = 1$. Let $\mathbf{A} \in \{0,1\}^{|\mathcal{V}| \times |\mathcal{V}|}$ be the adjacency matrix of the graph and $\mathbf{D}$ be the diagonal degree matrix where $\mathbf{D}_{ii} = \sum_j \mathbf{A}_{ij}$. The normalized adjacency matrix is defined as $\tilde{\mathbf{A}} = \mathbf{D}^{-\frac{1}{2}} \mathbf{A} \mathbf{D}^{-\frac{1}{2}}$. In the widely adopted LightGCN [21], the message passing mechanism in layer $l$ can be formally written as:

$$\mathbf{X}^{(l+1)} = \tilde{\mathbf{A}} \mathbf{X}^{(l)}, \tag{1}$$

where $\mathbf{X}^{(l)} \in \mathbb{R}^{|\mathcal{V}| \times d}$ is the node embedding matrix at layer $l$, with $d$ representing the embedding dimension. The recommendation score for user $u$ and item $i$ is then computed as the inner product between their respective embeddings: $y_{ui} = \mathbf{x}_u^T \mathbf{x}_i$.

### 2.2 Graph Laplacian and Dirichlet Energy

The graph Laplacian is a fundamental operator that captures the structural properties of a graph. The symmetrically normalized Laplacian [13, 59] is defined as:

$$\tilde{\mathbf{L}} = \mathbf{I} - \mathbf{D}^{-\frac{1}{2}} \mathbf{A} \mathbf{D}^{-\frac{1}{2}}. \tag{2}$$

Given node embeddings $\mathbf{X} \in \mathbb{R}^{|\mathcal{V}| \times d}$, the Dirichlet energy on the graph measures the smoothness of node embeddings with respect to the graph structure:

**Definition 1 (Graph Dirichlet Energy [13])** *The Dirichlet energy of node embeddings* $\mathbf{X}$ *with respect to graph* $\mathcal{G}$ *is defined as:*

$$E_{\mathcal{G}}(\mathbf{X}) := \mathrm{Tr}(\mathbf{X}^T \tilde{\mathbf{L}} \mathbf{X}) = \frac{1}{2} \sum_{(i,j) \in \mathcal{E}} \left\| \mathbf{x}_i / \sqrt{d_i} - \mathbf{x}_j / \sqrt{d_j} \right\|_2^2, \tag{3}$$

*where* $\mathrm{Tr}(\cdot)$ *denotes the matrix trace,* $\mathbf{x}_i \in \mathbb{R}^d$ *is the embedding of node* $i$, *and* $d_i$ *is its degree.*

Building on this global measure, we define a localized node Dirichlet energy:

**Definition 2 (Local Dirichlet Energy)** *The local Dirichlet energy of a node* $v$ *is defined as:*

$$E_v(\mathbf{x}_v) := \frac{1}{2} \sum_{j \in \mathcal{N}(v)} \left\| \mathbf{x}_v / \sqrt{d_v} - \mathbf{x}_j / \sqrt{d_j} \right\|_2^2, \tag{4}$$

*where* $\mathcal{N}(v)$ *denotes the set of neighbors of node* $v$.

### 2.3 Simplicial Complexes

Simplicial complexes (SC) provide a powerful framework to generalize graphs to higher-order relationships [77]. These structures are combinations of simplices, which we define as follows:

**Definition 3 ($k$-simplex)** *A* $k$-*simplex* $\sigma^k$ *is the convex hull of* $k + 1$ *affinely independent points* $S = \{v_0, v_1, \ldots, v_k\}$, *which constitute the vertices of the simplex. The dimension (or order) of the simplex is* $k$. *A* $j$-*simplex* ($j < k$) *formed by a subset of* $S$ *is called a* $j$-*face of* $\sigma^k$.

A $k$-simplex represents connections among $k + 1$ nodes: a 0-simplex is a node, a 1-simplex is an edge, a 2-simplex is a triangle, *etc*. These simplices are organized into simplicial complexes:

**Definition 4 (Simplicial Complex)** *A simplicial complex* $\mathcal{X}$ *is a finite collection of simplices satisfying the following properties: (1) Every face of a simplex in* $\mathcal{X}$ *is also in* $\mathcal{X}$. *(2) The non-empty intersection of any two simplices in* $\mathcal{X}$ *is a face of both.*

*The dimension (or order) of the simplicial complex* $\mathcal{X}$ *is the maximum dimension of any simplex in* $\mathcal{X}$.

The structure of a simplicial complex can be algebraically represented using boundary matrices:

**Definition 5 (Boundary Matrix)** *The boundary matrix* $\mathbf{B}_k$ *maps* $k$-*simplices to* $(k − 1)$-*simplices by encoding the incidence relations between them. Specifically,* $\mathbf{B}_k(i, j) = \pm 1$ *if the* $(k − 1)$-*simplex* $\sigma_i^{k-1}$ *is a face of the* $k$-*simplex* $\sigma_j^k$, *and 0 otherwise. The sign is determined by the orientation of simplices. Detailed discussion of orientations and examples can be found in Appendix D.*

Now we can define the Hodge Laplacians, generalizing the graph Laplacian to higher-order structures:

**Definition 6 (Hodge Laplacian [22])** *For a* $K$-*dimensional simplicial complex, the* $k$-*th Hodge Laplacian* $\mathbf{L}_k$ *is defined as:*

$$\mathbf{L}_0 = \mathbf{B}_1 \mathbf{B}_1^T, \quad \mathbf{L}_K = \mathbf{B}_K^T \mathbf{B}_K, \tag{5}$$

$$\mathbf{L}_k = \mathbf{B}_k^T \mathbf{B}_k + \mathbf{B}_{k+1} \mathbf{B}_{k+1}^T \quad (1 \leq k < K). \tag{6}$$

## 3 Topology Bias from Dirichlet Energy Perspective

In this section, we provide a theoretical analysis of how topology bias shapes the message passing process in graph-based recommender systems. Our analysis uncovers the mechanism by which popular nodes are systematically favored, a phenomenon rooted in an energy minimization dynamic.

### 3.1 Message Passing as Dirichlet Energy Optimization

We begin by establishing a fundamental equivalence between graph message passing and a Dirichlet energy minimization problem. This equivalence allows us to understand how the underlying graph topology directly governs information propagation.

**Lemma 3.1** *The message passing mechanism is equivalent to optimizing the following energy function with Tikhonov regularization [46] and initial embedding matrix* $\mathbf{X}^{(0)}$:

$$\min_{\mathbf{X}} \mathcal{J}(\mathbf{X}) = \frac{1}{2}\mathrm{Tr}(\mathbf{X}^T \tilde{\mathbf{L}} \mathbf{X}) + \frac{\mu}{2}\|\mathbf{X} - \mathbf{X}^{(0)}\|_2^2, \quad \mu > 0, \tag{7}$$

*where* $\tilde{\mathbf{L}}$ *is the normalized graph Laplacian, and* $\mu$ *is a regularization parameter.*

□ *Proof in Appendix A.1.*

Lemma 3.1 reveals that the message passing operation implicitly solves a global smoothness objective. Topology bias directly alters this energy landscape, thus shaping the resulting embeddings in a degree-dependent manner, as shown by the positive correlation in Figure 1(b).

## 3.2 Effect of Topology Bias on Embedding Norms

Next, we characterize how embeddings after message passing reflect the topology bias. Specifically, we show that nodes with higher degrees unavoidably accumulate larger embedding norms, a dynamic we refer to as *norm concentration*.

**Corollary 3.1** *(**Norm Concentration**) Let* $\mathbf{x}_v^*$ *denote the embedding of node* $v$ *in the minimizer* $\mathbf{X}^*$ *of Equation 7. Its squared norm is lower bounded by its degree* $d_v$:

$$\|\mathbf{x}_v^*\|_2^2 \geq d_v \cdot C(\tilde{\mathbf{L}}, \mathbf{X}^{(0)}), \tag{8}$$

*where* $C$ *depends on the graph Laplacian* $\tilde{\mathbf{L}}$ *and initial embedding* $\mathbf{X}^{(0)}$.

□ *Proof in Appendix A.2.*

Corollary 3.1 formalizes that popular nodes (with larger $d_v$) necessarily acquire larger norms during message passing, often resulting in higher recommendation scores. This effect is also noted by Kim et al. [26] in their empirical observations.

## 3.3 Topology Bias in Embedding Update Dynamics

We further show how topology bias persists and is amplified at every message passing step by bounding each node's embedding update in terms of its local graph energy.

**Lemma 3.2** *The squared norm of the embedding update for node* $v$ *in a single message passing layer is upper bounded as follows:*

$$\|\Delta\mathbf{x}_v\|_2^2 \leq E(v), \tag{9}$$

*where* $\Delta\mathbf{x}_v$ *denotes the change in embedding for node* $v$ *after message passing, and* $E(v)$ *is the local Dirichlet energy of* $v$, *as defined in Definition 2.*

□ *Proof in Appendix A.3.*

Lemma 3.2 exposes a self-reinforcing cycle: since the Dirichlet energy $E(v)$ itself is greater for higher-degree (popular) nodes as shown in Figure 1(b), these nodes receive larger embedding updates at each propagation step and potentially aggregate more information.

# 4 Methodology

## 4.1 Hodge Decomposition and Simplicial Dirichlet Energy

**Theorem 1 (Hodge Decomposition [22])** *The* $k$-*simplicial space* $\mathbb{R}^{N_k}$ *admits an orthogonal direct sum decomposition*

$$\mathbb{R}^{\mathcal{X}_k} = \mathrm{im}(\mathbf{B}_k^T) \oplus \ker(\mathbf{L}_k) \oplus \mathrm{im}(\mathbf{B}_{k+1}), \tag{10}$$

*where* $\mathrm{im}(\cdot)$ *and* $\ker(\cdot)$ *are shorthand for the image and kernel spaces of the matrices,* $\oplus$ *represents the union of orthogonal subspaces,* $N_k$ *is the cardinality of the space of* $k$-*simplex signals.*

Theorem 1 shows that any $k$-order simplicial signal $\mathbf{x}_{\sigma^k}$ can be decomposed [42]:

$$\mathbf{x}_{\sigma^k} = \mathbf{x}_{\sigma^k, G} + \mathbf{x}_{\sigma^k, H} + \mathbf{x}_{\sigma^k, C}, \tag{11}$$

where the gradient component $\mathbf{x}_{\sigma^k, G} = \mathbf{B}_k^T \mathbf{x}_{\sigma^{k-1}}$ (corresponding to Equation 20), curl component $\mathbf{x}_{\sigma^k, C} = \mathbf{B}_{k+1} \mathbf{x}_{\sigma^{k+1}}$ (corresponding to Equation 22), and the harmonic component $\mathbf{x}_{\sigma^k, H}$ satisfies

$\mathbf{L}_k \mathbf{x}_{\sigma^k, H} = \mathbf{0}$. This decomposition provides theoretical guidance for the interaction and conversion between simplicial signals of different orders.

**Definition 7 (Simplicial Dirichlet Energy [61])** *The Dirichlet energy of simplices of order $k$ can be defined as:*

$$E_{\sigma^k}(\mathbf{X}_{\sigma_i^k}) := \left\| \mathbf{B}_k \mathbf{X}_{\sigma^k} \right\|_2^2 + \left\| \mathbf{B}_{k+1}^T \mathbf{X}_{\sigma^k} \right\|_2^2 \tag{12}$$

$$= \underbrace{\sum_{\sigma_i^k \cap \sigma_j^k \neq \emptyset} \left\| \mathbf{x}_{\sigma_i^k} - \mathbf{x}_{\sigma_j^k} \right\|_2^2}_{shared\ lower\ simplices} + \underbrace{\sum_{\sigma_i^k, \sigma_j^k \subset \sigma^{k+1}} \left\| \mathbf{x}_{\sigma_i^k} - \mathbf{x}_{\sigma_j^k} \right\|_2^2}_{shared\ upper\ simplices}. \tag{13}$$

Different from the Dirichlet energy defined on a graph, the simplicial Dirichlet energy consists of two terms: energy between simplices sharing the same lower order component simplices (lower simplices) and energy between simplices consisting of the same higher order simplices (higher simplices).

Consider optimizing the simplicial Dirichlet energy similar to Equation 7:

$$\min_{\mathbf{X}_{\sigma^k}} \quad \frac{1}{2} E_{\sigma^k}(\mathbf{X}_{\sigma_i^k}) + \frac{\mu}{2} \left\| \mathbf{X}_{\sigma^k} - \mathbf{X}_{\sigma^k}^{(0)} \right\|_2^2. \tag{14}$$

The gradient with respect to $\mathbf{X}_{\sigma^k}$ is:

$$\nabla_{\mathbf{X}_{\sigma^k}} = \mathbf{B}_k^T \mathbf{B}_k \mathbf{X}_{\sigma^k} + \mathbf{B}_{k+1} \mathbf{B}_{k+1}^T \mathbf{X}_{\sigma^k} + \mu(\mathbf{X}_{\sigma^k} - \mathbf{X}_{\sigma^k}^{(0)}) = \mathbf{L}_k \mathbf{X}_{\sigma^k} + \mu(\mathbf{X}_{\sigma^k} - \mathbf{X}_{\sigma^k}^{(0)}). \tag{15}$$

One gradient descent step with learning rate $\eta$ can be written as:

$$\mathbf{X}_{\sigma^k}^{(1)} = \mathbf{X}_{\sigma^k}^{(0)} - \eta \nabla_{\mathbf{X}_{\sigma^k}} = \mathbf{X}_{\sigma^k}^{(0)} - \eta \mathbf{L}_k \mathbf{X}_{\sigma^k}^{(0)} = (\mathbf{I} - \eta \mathbf{L}_k) \mathbf{X}_{\sigma^k}^{(0)}. \tag{16}$$

Notably, Equation 16 is the simplicial version of graph message passing. The orthonormal eigen-decomposition of $\mathbf{L}_k$ can be written as:

$$\mathbf{L}_k = \sum_{i=1}^N \lambda_i \mathbf{u}_i \mathbf{u}_i^T, \tag{17}$$

where $0 = \lambda_1 = \ldots = \lambda_r < \lambda_{r+1} < \ldots \leq \lambda_N$ are the eigenvalues and $\mathbf{u}_i$ are the corresponding eigenvectors. Similarly, we can define a resolvent to this optimization problem as in Equation 31:

$$\mathbf{R}_k = \mu(\mathbf{L}_k + \mu \mathbf{I})^{-1}. \tag{18}$$

For every initial embedding vector $\mathbf{v}$, $\mathbf{R}_k \mathbf{v}$ yields the corresponding energy minimizer. Now we show how simplicial message passing can alleviate the topology bias.

**Lemma 4.1** *For any $\mathbf{v} \in \ker(\mathbf{B}_{k+1})$ we have:* $\|\mathbf{R}_k \mathbf{v}\|_2^2 < \|\mathbf{v}\|_2^2$.

□ *Proof of Lemma 4.1 is in Appendix A.4.*

Lemma 4.1 reveals that in simplicial message passing, for any embedding in the kernel of the upper adjacency operator ($\ker(\mathbf{B}_{k+1})$), the norm is strictly reduced by the resolvent operator $\mathbf{R}_k$. This means that, unlike traditional graph-based propagation which tends to inflate the norms of embeddings for popular (high-degree) nodes, simplicial message passing can contract such dominant embeddings. By shrinking oversized embedding norms where bias accumulates, the simplicial approach helps regularize the embedding space, leading to a more balanced embedding landscape.

## 4.2 The Proposed TSP

Guided by our theoretical analysis, we propose **Test-time Simplicial Propagation (TSP)** based on simplicial Dirichlet energy optimization, extending the message passing procedure to simplicial complexes. TSP comprises the following key modules: **Semantic Graph Construction**, **Intra-Simplex Smoothing**, **Inter-Simplex Propagation**, and **Multi-Order Fusion**. An overview of the whole pipeline is in Figure 2. We also provide an algorithm description in pseudo code in Appendix E.

**Semantic Graph Construction** The objective of this module is to furnish a topology that is more reflective of *semantic* relationships between nodes, rather than relying solely on raw interaction patterns. Formally, given pretrained node embedding $\mathbf{X}$ and hyperparameter $\theta$ controlling the

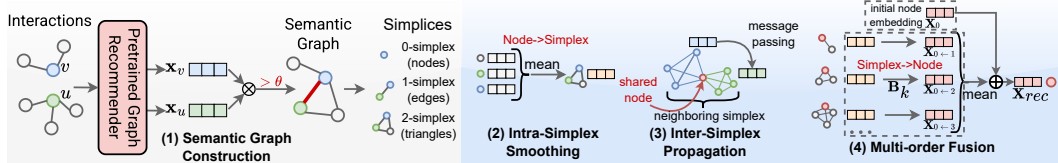

Figure 2: The pipeline of our proposed TSP, consisting of four main steps: (1) Semantic Graph Construction, which builds a semantic graph based on node embeddings; (2) Intra-Simplex Smoothing, which aggregates information within each simplex; (3) Inter-Simplex Propagation, which performs message passing between simplices; and (4) Multi-Order Fusion, which combines embeddings from different orders of simplices for the final recommendation.

similarity threshold, we can construct the adjacency matrix of the semantic graph $\mathbf{A}_S$:

$$\mathbf{A}_S[i,j] = \begin{cases} 1, & \mathbf{x}_i^T \mathbf{x}_j >= \theta \\ 0, & \text{otherwise} \end{cases}, \tag{19}$$

where $\mathbf{x}_i$ is the embedding of node $i$, $\mathbf{A}_S[i,j]$ means the element at the $i$-th row, $j$-th column in $\mathbf{A}_S$. This process naturally densifies the tail nodes by giving low-degree nodes additional connections, thus partially alleviating the topology bias [4, 69]. Then we lift this semantic graph into simplicial complex, as described in Appendix I.

**Intra-Simplex Smoothing**    Once we have $\mathbf{A}_S$, we lift it into a simplicial complex by incorporating nodes, edges, and higher-order simplices (triangles, tetrahedrons, etc.) that capture multi-node connectivity. Each $k$-simplex $\sigma^k$ consists of $k+1$ nodes joined by semantic adjacency. Let $\mathbf{B}_k$ be the boundary matrix of $k$-simplices. These matrices enable us to convert node embeddings to high-order simplices signals as in Equation 11. Formally, we can aggregate the information for $\sigma^k$ from its component nodes as follows:

$$\left(\mathbf{B}_k^T \mathbf{B}_{k-1}^T \ldots \mathbf{B}_1^T\right)\mathbf{X} \tag{20}$$

where $\mathbf{X}_{\sigma^k}$ is the embedding matrix of $k$-simplices. This process performs smoothing inside the simplex, and also provides initial embeddings for each simplex without the need for embedding learning. The factor $\prod_{i=1}^{k}\mathbf{B_k}$ converts the embeddings of component nodes into simplex embeddings.

**Inter-Simplex Propagation**    Although intra-simplex averaging helps to smooth embeddings within each simplex structure, it alone does not capture the relations *between* simplices. To address this, we introduce *inter-simplex propagation*. More specifically, we utilize the Hodge Laplacians in Equation 6 to define the message passing between simplices within each order $k$:

$$\mathbf{X}_{\sigma^k}^{(l+1)} = \left(\mathbf{I} - \beta \mathbf{L}_k\right)\mathbf{X}_{\sigma^k}^{(l)}, \tag{21}$$

where $\mathbf{L}_k$ is the Hodge Laplacian of order $k$, $l$ is the message passing layer, and $\beta$ controls the signal filtering on simplices [62]. This message passing process further refine the representation based on higher-order simplicial structure, incorporating simplex interactions.

**Multi-Order Fusion**    To obtain the final recommendation scores, we need to fuse the embeddings from different orders of simplices. We can perform the reverse process of Equation 20 to fuse the embeddings of all the simplices into node embeddings:

$$\mathbf{X}_{0 \leftarrow k} = \left(\mathbf{B}_1 \mathbf{B}_2 \ldots \mathbf{B}_k\right) X_{\sigma^k}^{(L)} \tag{22}$$

where $\mathbf{X}_{0 \leftarrow k}$ represents the node embeddings converted from $k$-simplices. We then produce the final representation for recommendation by performing mean pooling of multi-order representations:

$$\mathbf{X}_{\text{rec}} = \mathbf{X}_0 + \text{Mean}\left(\mathbf{X}_{0 \leftarrow 1}, \ldots, \mathbf{X}_{0 \leftarrow K}\right), \tag{23}$$

where $\mathbf{X}_0$ is the original pretrained embedding, and $\mathbf{X}_{0 \leftarrow k}$ are the embeddings from $k$-order simplices after message passing. This multi-order fusion ensures that our final embeddings incorporate both interaction information and higher-order semantic relations.

# 5 Experiments

In this section, we empirically evaluate the effectiveness of our proposed method TSP. We aim to answer the following research questions:

- **RQ1:** How does TSP perform compared to existing state-of-the-art debiasing methods?
- **RQ2:** Can TSP improve recommendation fairness and diversity?
- **RQ3:** Can TSP produce fairer item embeddings, as suggested by the theoretical analysis?
- **RQ4:** What is the contribution of each component in TSP to model performance?
- **RQ5:** How much computational overhead is introduced by TSP?

## 5.1 Experimental Setup

**Datasets** We conduct experiments on five real-world datasets representing diverse domains: Yelp2018 (business reviews), Gowalla (location check-ins), Adressa (news), Globo (news), and ML10M (movie ratings). User ratings in ML10M are converted to implicit feedback (1 if rated, 0 otherwise). Details of these dataset statistics are in Appendix F.

**Evaluation Protocols** We split the datasets into 80%, 10%, 10% as train, validation, and test sets following the unbiased sampling method proposed by Wei et al. [56]. We report Recall@20 (R@20) and NDCG@20 (N@20) averaged across all users. Specifically, we also report performance of the 20% least popular items (tail items) beside overall performance. Evaluation details are in Appendix B.

**Backbone Models** We demonstrate TSP's effectiveness by applying it to three strong GNN-based backbone models: **LightGCN** [21] is a simplified GCN model widely used for collaborative filtering. **SimGCL** [65] is a graph contrastive learning method using noise injection for augmentation. **Light-GCL** [9] is the state-of-the-art contrastive learning model using SVD for graph view construction.

**Debiasing Baselines** We compare TSP against several representative debiasing methods: **IPS** [43] uses inverse propensity scoring to reweight interactions based on item popularity. **CausE** [6] relies on causal inference to learn representations robust to exposure bias. **MACR** [56] employs counterfactual reasoning based on a causal graph to mitigate popularity bias. **SDRO** [58] uses distributionally robust optimization to improve worst-case performance.

**Implementation** GNNs use 3 message passing layers and an embedding dimension of 64. We use the Adam optimizer with learning rate 0.001, weight decay 1e-4, batch size 4096. Training runs for up to 500 epochs with early stopping patience 50. Further details can be found in Appendix G.

## 5.2 Recommendation Performance (RQ1)

Table 1 presents the recommendation performance specifically for tail items, while Figure 3 shows the overall performance across all items on Gowalla dataset. More empirical results of overall recommendation performance can be found in Appendix C.

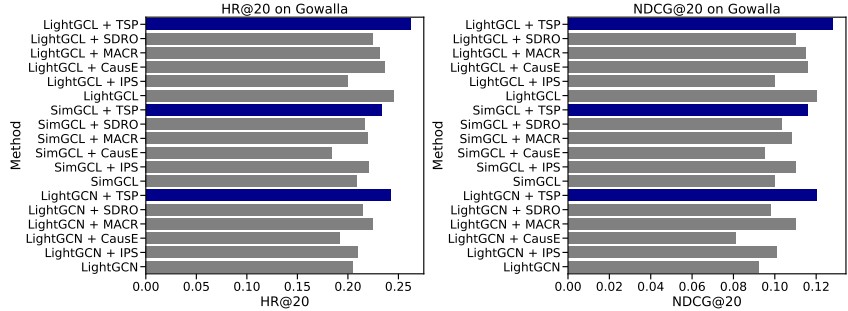

Figure 3: The overall performance of TSP and other baselines on Gowalla dataset.

We have following key observations: (1) TSP consistently achieves substantial improvements in Recall@20 and NDCG@20 for tail items across all datasets and backbone models (Table 1). It frequently outperforms the best baseline debiasing method by a significant margin (%Improve). This validates the strong ability of TSP to improve tail performance. (2) While the primary goal is mitigating topology bias, TSP often maintains or even improves overall performance compared to

Table 1: Performance on tail items (20% least popular items). Best results are in **bold**, and second-best results are underlined. %Improve is TSP's relative gain over the second-best method. * indicates the statistically significant improvements (*i.e.,* two-sided t-test with p<0.05) over the best baseline.

| | Adressa | | Gowalla | | Yelp2018 | | ML10M | | Globo | |
|---|---|---|---|---|---|---|---|---|---|---|
| Method | R@20 | N@20 | R@20 | N@20 | R@20 | N@20 | R@20 | N@20 | R@20 | N@20 |
| LightGCN | 0.015 | 0.007 | 0.006 | 0.006 | 0.002 | 0.001 | 0.002 | 0.001 | 0.001 | 0.002 |
| + IPS | 0.040 | 0.020 | 0.030 | **0.025** | 0.006 | 0.007 | 0.011 | 0.005 | 0.022 | 0.012 |
| + CausE | 0.023 | 0.011 | 0.020 | 0.013 | 0.007 | 0.004 | 0.007 | 0.002 | 0.014 | 0.008 |
| + MACR | 0.016 | 0.007 | 0.014 | 0.008 | 0.005 | 0.004 | 0.007 | 0.002 | 0.006 | 0.005 |
| + SDRO | 0.014 | 0.007 | 0.012 | 0.007 | 0.005 | 0.003 | 0.007 | 0.002 | 0.003 | 0.004 |
| + TSP (Ours) | **0.055** | **0.024** | **0.038** | 0.021 | **0.009** | **0.008** | **0.013** | **0.008** | **0.024** | **0.013** |
| *%Improve* | *37.50** | *20.00** | *26.67** | *-* | *28.57** | *14.29** | *18.18** | *60.00** | *9.09** | *8.33** |
| SimGCL | 0.016 | 0.007 | 0.008 | 0.007 | 0.002 | 0.003 | 0.003 | 0.002 | 0.002 | 0.001 |
| + IPS | 0.052 | 0.027 | 0.021 | 0.018 | 0.008 | 0.006 | 0.012 | 0.008 | **0.025** | 0.011 |
| + CausE | 0.018 | 0.008 | 0.028 | 0.018 | 0.006 | 0.005 | 0.010 | 0.004 | 0.010 | 0.008 |
| + MACR | 0.014 | 0.006 | 0.017 | 0.006 | 0.005 | 0.004 | 0.013 | 0.009 | 0.012 | 0.008 |
| + SDRO | 0.016 | 0.007 | 0.014 | 0.007 | 0.006 | 0.003 | 0.008 | 0.003 | 0.007 | 0.006 |
| + TSP (Ours) | **0.085** | **0.046** | **0.055** | **0.047** | **0.015** | **0.014** | **0.016** | **0.012** | **0.025** | **0.017** |
| *%Improve* | *63.46** | *70.37** | *96.43** | *161.11** | *87.50** | *133.33** | *23.08** | *23.33** | *-* | *54.55** |
| LightGCN | 0.020 | 0.009 | 0.010 | 0.008 | 0.003 | 0.003 | 0.003 | 0.002 | 0.004 | 0.004 |
| + IPS | 0.045 | 0.022 | 0.040 | 0.030 | 0.010 | 0.010 | 0.010 | 0.006 | 0.024 | 0.016 |
| + CausE | 0.022 | 0.008 | 0.026 | 0.019 | 0.002 | 0.002 | 0.003 | 0.001 | 0.015 | 0.011 |
| + MACR | 0.008 | 0.005 | 0.015 | 0.012 | 0.011 | 0.012 | 0.011 | 0.007 | 0.013 | 0.007 |
| + SDRO | 0.015 | 0.007 | 0.013 | 0.007 | 0.010 | 0.003 | 0.007 | 0.002 | 0.007 | 0.005 |
| + TSP (Ours) | **0.075** | **0.047** | **0.050** | **0.032** | **0.020** | **0.019** | **0.014** | **0.010** | **0.032** | **0.018** |
| *%Improve* | *66.67** | *113.64** | *25.00** | *6.67** | *81.82** | *58.33** | *27.27** | *66.67** | *133.33** | *12.50** |

the backbone (Figure 3, Appendix C). In many cases, it also outperforms other debiasing methods on overall metrics, suggesting TSP can effectively balance representations without sacrificing overall recommendation quality. (3) The improvements offered by TSP are particularly pronounced when applied to contrastive learning backbones (SimGCL, LightGCL) compared to LightGCN. This might indicate that contrastive methods can benefit more from higher-order structures.

## 5.3 Recommendation Fairness and Diversity (RQ2)

To assess the effect of TSP on recommendation fairness and diversity, we evaluate two established metrics, Expected Free Discovery (EFD) [47] and Average Percentage of Tail Items (APT) [2], with detailed definitions provided in Appendix B. As reported in Table 2, TSP consistently outperforms competitive baselines on both EFD@20 and APT@20 across all datasets when using the LightGCN backbone, indicating that our method improves the exposure of long-tail content while enhancing overall fairness.

Table 2: Performance on LightGCN backbone with respect to EFD@20 and APT@20. Best results are in **bold**, and second-best results are underlined.

| | Adressa | | Gowalla | | Yelp2018 | | ML10M | | Globo | |
|---|---|---|---|---|---|---|---|---|---|---|
| Method | EFD@20 | APT@20 | EFD@20 | APT@20 | EFD@20 | APT@20 | EFD@20 | APT@20 | EFD@20 | APT@20 |
| LightGCN | 0.823 | 0.124 | 0.791 | 0.131 | 0.752 | 0.085 | 0.715 | 0.072 | 0.724 | 0.064 |
| + IPS | 0.831 | 0.133 | 0.801 | 0.156 | 0.764 | 0.101 | 0.732 | 0.080 | 0.714 | 0.073 |
| + CausE | 0.846 | 0.158 | 0.831 | 0.157 | 0.793 | 0.105 | 0.763 | 0.093 | 0.735 | 0.079 |
| + MACR | 0.901 | 0.147 | 0.856 | 0.150 | **0.814** | 0.118 | 0.794 | 0.107 | 0.756 | 0.121 |
| + SDRO | 0.832 | 0.136 | 0.781 | 0.138 | 0.735 | 0.110 | 0.705 | 0.097 | 0.721 | 0.112 |
| + TSP (Ours) | **0.923** | **0.217** | **0.875** | **0.164** | 0.813 | **0.133** | **0.802** | **0.116** | **0.773** | **0.149** |

## 5.4 Distribution of Item Embeddings (RQ3)

To explore the impact of our proposed method on the distribution of learned embeddings, we project the item embeddings learned by LightGCN and LightGCL on the Gowalla dataset into a two-dimensional space using t-SNE visualization. We compare these results with competitive MACR baselines, as shown in Figure 4. Our key observations are as follows: (1) LightGCL exhibits a more balanced distribution of item embeddings compared to LightGCN, aligning with the expectation that contrastive learning methods can learn more uniform representations [51]. (2) The incorporation of simplicial propagation in our method results in a more uniform distribution of item embeddings, validating the effectiveness of simplicial message passing.

Table 3: Ablation study on Gowalla dataset. We report the overall and 20% tail item results of Top 20 performance.

| Variant | Overall | | Tail | |
|---|---|---|---|---|
| | R@20 | N@20 | R@20 | N@20 |
| TSP-Full | 0.082 | 0.058 | 0.038 | 0.021 |
| TSP-SE | 0.056 | 0.041 | 0.017 | 0.009 |
| LGN | 0.047 | 0.038 | 0.006 | 0.006 |

Table 4: TSP preprocessing and inference time costs compared with LightGCN backbone training/inference time.

| Time Cost | Adressa | Gowalla | Yelp18 | ML10M | Globo |
|---|---|---|---|---|---|
| LGN Train | 10min | 35min | 1h40m | 9h | 3h |
| TSP Preproc | 2.54s | 15.97s | 26.32s | 91.34s | 44.29s |
| LGN Infer | 0.037s | 0.080s | 0.134s | 0.575s | 0.501s |
| TSP Infer | 0.098s | 0.130s | 0.200s | 0.650s | 0.606s |

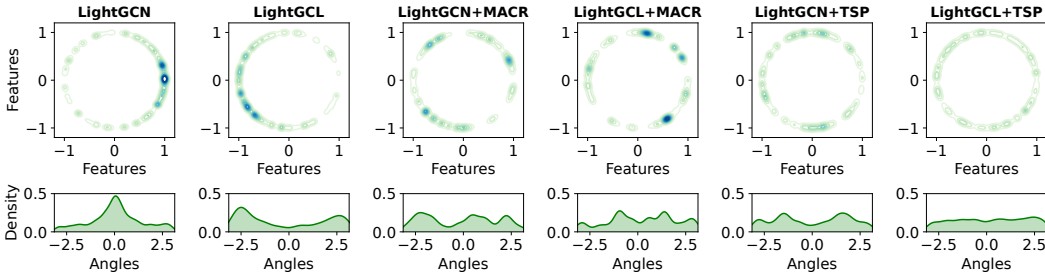

Figure 4: t-SNE visualization of item embeddings learned on Gowalla dataset. The top figures plot the Gaussian KDE of embeddings projected to $\mathbb{R}^2$. The darker the color, the more items are in this region. The figures below show the KDE of angles (i.e. $arctan2(y, x)$ for each point $(x, y)$).

## 5.5 Ablation Study (RQ4)

To isolate the contribution of simplicial message passing and semantic graph construction to the final performance gains, we conduct an ablation study on Gowalla dataset. The results are in Table 3, with more results in Appendix H. Specifically, we compare following variants of TSP with same LightGCN backbone: (1) **TSP-Full**, the original model, (2) **TSP-SE**, which uses the same semantic graph as TSP, but only performs graph message passing, (3) **LGN**, the LightGCN baseline. The results show that while TSP-SE does provide some improvement, suggesting that denoising the graph structure can improve model performance. The full TSP framework, incorporating message passing on the simplicial complex structure, yields significantly larger gains. This confirms that explicitly modeling and propagating information through higher-order structures is crucial for effectively smoothing representations and achieving fairer recommendations.

## 5.6 Scalability (RQ5)

To assess practical viability (RQ5), we measure the preprocessing time (building the complex and Laplacian) and the additional inference time introduced by TSP compared to the LightGCN backbone. Table 4 shows these times across datasets. The lightweight one-time preprocessing and efficient inference compared to the backbone LightGCN suggests that our approach introduces minimal additional computational overhead. Notably, as the dataset size increases, the relative computational overhead introduced by TSP diminishes in comparison to the backbone model.

## 6 Related Works

**Graph Neural Networks for Recommendation**    Graph Neural Networks (GNNs) [15, 48] are widely used for modeling high-order interactions in recommendation tasks. Early works like NGCF [52] utilize graph convolutions to iteratively aggregate collaborative signals. Recent advances focus on improving GNN architectures to enhance efficiency and accuracy, such as LightGCN [21], UltraGCN [37], and SVD-GCN [40]. The integration of contrastive learning [60, 65, 70, 49] has further improved representation robustness, with models like SimGCL [65] demonstrating the benefits of self-supervised learning objectives and lightweight graph augmentations. Emerging research also explores the utilization of large language models (LLMs) enhancing graph recommenders [24, 57, 54], expanding the applicability of graph-based methods.

**Higher-Order and Simplicial Neural Networks**    While GNNs effectively capture pairwise relationships, their expressive power is limited by the 1-Weisfeiler-Lehman (1-WL) test [53], impeding their ability to model higher-order structures such as triangles, cliques, and communities [12]. To overcome this, recent studies employ simplicial complexes and higher-order message passing. Models like Simplicial Neural Networks (SNN) [17], BSCNet [11], and HSN [20] extend convolutions beyond simple edges, leveraging Hodge Laplacians for richer structural representations. Simplicial attention mechanisms [19, 18, 29] are also developed to model interactions among higher-order simplices.

**Popularity Bias Mitigation**    Popularity bias remains a critical challenge for fair recommendation. Earlier approaches utilize regularization [7, 50] or adversarial frameworks [33, 68, 27] to reduce popularity-driven ranking bias. Causal modeling has emerged as a powerful tool to disentangle popularity effects [56, 75], with counterfactual reasoning being applied to mitigate bias at the inference stage. Newer perspectives focus on employing topological interventions or stochastic graph augmentations [36, 73]. Recent research also investigates debiasing via distributionally robust optimization [58], embedding landscape modeling [32, 28], and reinforcement learning frameworks [44, 30] to achieve a balance between recommendation quality and fairness.

**Oversmoothing in Graph Neural Networks**    The performance of GNNs is strongly affected by the depth of the network [5, 14, 39]. As the number of layers increases, the node embeddings contract and converge exponentially toward a constant vector, severely degrading the predictive accuracy [25]. Several metrics have been proposed to quantify oversmoothing and characterize message-passing dynamics, including Dirichlet energy and Mean Average Distance (MAD). Although we also employ Dirichlet energy to analyze message passing, our focus is fundamentally different: we investigate the topology bias instead of network depth. Specifically, we provide the first analysis of how graph topology influences Dirichlet energy and extend this energy-based characterization to higher-order simplicial complexes.

## 7    Conclusion

This paper addresses topology bias in graph-based recommender systems by analyzing message passing through a Dirichlet energy perspective. Based on our theoretical analysis, we propose Test-time Simplicial Propagation (TSP) to address this bias by using higher-order simplicial complex structures for more balanced message passing. Extensive experiments demonstrate that TSP effectively reduces the topology bias and improves recommendation quality.

## Acknowlegments

This paper is supported by NSFC (No. 62176155), Shanghai Municipal Science and Technology Major Project, China, under grant No. 2021SHZDZX0102.

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

# Appendix

## A Proofs

### A.1 Proof of Lemma 3.1

**Lemma 3.1** *The message passing mechanism is equivalent to optimizing the following energy function with Tikhonov regularization [46] and initial embedding matrix $\mathbf{X}^{(0)}$:*

$$\min_{\mathbf{X}} \mathcal{J}(\mathbf{X}) = \frac{1}{2}\mathrm{Tr}(\mathbf{X}^T\tilde{\mathbf{L}}\mathbf{X}) + \frac{\mu}{2}\|\mathbf{X} - \mathbf{X}^{(0)}\|_2^2, \quad \mu > 0, \tag{7}$$

*where $\tilde{\mathbf{L}}$ is the normalized graph Laplacian, and $\mu$ is a regularization parameter.*

**Proof** *The gradient of the objective function in Equation (7) with respect to $\mathbf{X}$ is:*

$$\nabla_{\mathbf{X}}\mathcal{J}(\mathbf{X}) = \tilde{\mathbf{L}}\mathbf{X} + \mu(\mathbf{X} - \mathbf{X}^{(0)}) \tag{24}$$

*With a learning rate of $\eta$, one step of gradient descent yields:*

$$\mathbf{X}^{(1)} = \mathbf{X}^{(0)} - \eta\nabla_{\mathbf{X}}\mathcal{J}(\mathbf{X}^{(0)}) \tag{25}$$

$$= \mathbf{X}^{(0)} - \eta[\tilde{\mathbf{L}}\mathbf{X}^{(0)} + \mu(\mathbf{X}^{(0)} - \mathbf{X}^{(0)})] \tag{26}$$

$$= \mathbf{X}^{(0)} - \eta\tilde{\mathbf{L}}\mathbf{X}^{(0)} \tag{27}$$

$$= (\mathbf{I} - \eta\tilde{\mathbf{L}})\mathbf{X}^{(0)} \tag{28}$$

*When $\eta = 1$, Equation (28) becomes $\mathbf{X}^{(1)} = (\mathbf{I} - \tilde{\mathbf{L}})\mathbf{X}^{(0)} = \tilde{\mathbf{A}}\mathbf{X}^{(0)}$, which is exactly the message passing equation described in Equation (1). For $\eta < 1$, the operation corresponds to a low-pass filtering on graph signals, consistent with prior work [67].*

*Alternatively, we can derive the closed-form solution by setting the gradient in Equation (24) to zero:*

$$\tilde{\mathbf{L}}\mathbf{X}^* + \mu(\mathbf{X}^* - \mathbf{X}^{(0)}) = 0 \tag{29}$$

$$\Rightarrow (\tilde{\mathbf{L}} + \mu\mathbf{I})\mathbf{X}^* = \mu\mathbf{X}^{(0)} \tag{30}$$

$$\Rightarrow \mathbf{X}^* = \mu(\tilde{\mathbf{L}} + \mu\mathbf{I})^{-1}\mathbf{X}^{(0)} \tag{31}$$

*Let $\tilde{\mathbf{L}} = \mathbf{U}\boldsymbol{\Lambda}\mathbf{U}^T$ be the eigendecomposition of $\tilde{\mathbf{L}}$, where $\boldsymbol{\Lambda} = \mathrm{diag}(\lambda_1, \lambda_2, \ldots, \lambda_N)$ is the diagonal matrix of eigenvalues. Then:*

$$\mathbf{X}^* = \mu(\mathbf{U}\boldsymbol{\Lambda}\mathbf{U}^T + \mu\mathbf{I})^{-1}\mathbf{X}^{(0)} \tag{32}$$

$$= \mu\mathbf{U}(\boldsymbol{\Lambda} + \mu\mathbf{I})^{-1}\mathbf{U}^T\mathbf{X}^{(0)} \tag{33}$$

$$= \mathbf{U}\tilde{\boldsymbol{\Lambda}}\mathbf{U}^T\mathbf{X}^{(0)} \tag{34}$$

*where*

$$\tilde{\boldsymbol{\Lambda}} = \mu(\boldsymbol{\Lambda} + \mu\mathbf{I})^{-1} = \mathrm{diag}\left(\frac{\mu}{\lambda_k + \mu}\right) \quad \textit{for } 1 \leq k \leq N \tag{35}$$

*The embedding for node $v$ can then be expressed as:*

$$\mathbf{x}_v^* = \sum_{k=1}^{N}\left(\frac{\mu}{\lambda_k + \mu}\right)\mathbf{u}_k[v]\mathbf{c}_k, \tag{36}$$

*where $\mathbf{c}_k = \mathbf{u}_k^T\mathbf{X}^{(0)}$ and $\mathbf{u}_k[v]$ is the $v$-th element of eigenvector $\mathbf{u}_k$. The spectral filter $\frac{\mu}{\lambda_k + \mu}$ functions as a low-pass filter on graph signals, attenuating components corresponding to larger eigenvalues.*

## A.2 Proof of Corollary 3.1

**Corollary 3.1** (*Norm Concentration*) *Let $\mathbf{x}_v^*$ denote the embedding of node $v$ in the minimizer $\mathbf{X}^*$ of Equation 7. Its squared norm is lower bounded by its degree $d_v$:*

$$\|\mathbf{x}_v^*\|_2^2 \geq d_v \cdot C(\tilde{\mathbf{L}}, \mathbf{X}^{(0)}), \tag{8}$$

*where $C$ depends on the graph Laplacian $\tilde{\mathbf{L}}$ and initial embedding $\mathbf{X}^{(0)}$.*

**Proof** *For the node embedding described in Equation 36, we can obtain a lower bound of norm by only considering the smallest eigenvalue $\lambda_1 = 0$ corresponding to eigenvector $\mathbf{u}_1 = (\sqrt{d_1}, \sqrt{d_2}, \ldots, \sqrt{d_N})$, where $d_i$ is the degree of node $i$.*

$$\|\mathbf{x}_v^*\|_2^2 = \left\| \sum_{k=1}^N (\mu/(\tilde{\lambda}_k + \mu))\mathbf{u}_k[v]\mathbf{c}_k^T \right\|_2^2 \tag{37}$$

$$\geq \left\| \mu/(\tilde{\lambda}_1 + \mu))\mathbf{u}_1[v]\mathbf{c}_k^T \right\|_2^2 \tag{38}$$

$$= \|\mathbf{u}_1[v]\|_2^2 \cdot C(\tilde{\mathbf{L}}, \mathbf{X}^{(0)}) = d_v \cdot C(\tilde{\mathbf{L}}, \mathbf{X}^{(0)}) \tag{39}$$

## A.3 Proof of Lemma 3.2

**Lemma 3.2** *The squared norm of the embedding update for node $v$ in a single message passing layer is upper bounded as follows:*

$$\|\Delta\mathbf{x}_v\|_2^2 \leq E(v), \tag{9}$$

*where $\Delta\mathbf{x}_v$ denotes the change in embedding for node $v$ after message passing, and $E(v)$ is the local Dirichlet energy of $v$, as defined in Definition 2.*

**Proof** *We can write Equation 1 as:*

$$\Delta\mathbf{x}_v = \mathbf{x}_v^{(l)} - \mathbf{x}_v^{(l-1)} = \sum_{j \in \mathcal{N}(v)} \frac{1}{\sqrt{d_v d_j}}\mathbf{x}_j^{(l-1)} - \mathbf{x}_v^{(l-1)} \tag{40}$$

$$= \frac{1}{\sqrt{d_v}} \sum_{j \in \mathcal{N}(v)} \left( \frac{1}{\sqrt{d_j}}\mathbf{x}_j^{(l-1)} - \frac{1}{\sqrt{d_v}}\mathbf{x}_v^{(l-1)} \right). \tag{41}$$

$$\tag{42}$$

*Then we have:*

$$\|\Delta\mathbf{x}_v\|_2^2 = \frac{1}{d_v} \left\| \sum_{j \in \mathcal{N}_k(v)} \left( \frac{1}{\sqrt{d_j}}\mathbf{x}_j^{(l-1)} - \frac{1}{\sqrt{d_v}}\mathbf{x}_v^{(l-1)} \right) \right\|_2^2 \tag{43}$$

$$\leq \frac{1}{d_v} \cdot d_v \sum_{j \in \mathcal{N}_k(v)} \left\| \frac{1}{\sqrt{d_j}}\mathbf{x}_j^{(l-1)} - \frac{1}{\sqrt{d_v}}\mathbf{x}_v^{(l-1)} \right\|_2^2 \quad \textit{(Cauchy–Schwarz Inequality)} \tag{44}$$

$$= E(v) \tag{45}$$

## A.4 Proof of Lemma 4.1

**Lemma 4.1** *For any $\mathbf{v} \in \ker(\mathbf{B}_{k+1})$ we have: $\|\mathbf{R}_k\mathbf{v}\|_2^2 < \|\mathbf{v}\|_2^2$.*

**Proof**

$$\|\mathbf{R}_k\mathbf{v}\|_2^2 = \left( \sum_{i=1}^N \frac{\mu}{\lambda_i + \mu}\mathbf{u}_i\mathbf{u}_i^T\mathbf{v} \right)^2 \leq \left( \frac{\mu}{\lambda_{r+1} + \mu} \right)^2 \sum_{i=1}^N \underbrace{\|\mathbf{u_i}\|_2^2}_{1}(\mathbf{u}_i^T\mathbf{v})^2 \tag{46}$$

*And since $\mathbf{u}_i$ span the kernel of $\mathbf{B}_{k+1}$, we have*

$$\sum_{i=1}^N (\mathbf{u}_i^T\mathbf{v})^2 = \|\mathbf{v}\|_2^2. \tag{47}$$

*So we get*

$$\|\mathbf{R}_k\mathbf{v}\|_2^2 \leq \left(\frac{\mu}{\lambda_{r+1}+\mu}\right)^2 \|\mathbf{v}\|_2^2 < \|\mathbf{v}\|_2^2. \tag{48}$$

# B  Evaluation Protocols

## B.1  Unbiased Sampling

To fairly evaluate debiasing performance, standard random train-test splits are inadequate as they inherit the original data's popularity bias. We adopt the unbiased evaluation protocol from Wei et al. [56]. Specifically, we uniformly sample 10% of interactions for the test set and 10% for validation, ensuring all items have an equal chance of being sampled, regardless of their popularity. The remaining 80% form the training set. This creates unbiased test and validation sets with balanced popularity distributions, allowing for accurate assessment of long-tail performance.

## B.2  Evaluation Metrics

**Recall**  Recall@k is the proportion of relevant items found in the top-k recommendations. A higher Recall@k value indicates that the system is better at identifying and ranking relevant items for users. Formally, Recall@k is defined as:

$$\text{Recall@k} = \frac{\text{Number of Relevant Items in Top-k}}{\text{Total Number of Relevant Items}}. \tag{49}$$

**Normalized Discounted Cumulative Gain (NDCG) [23]**  NDCG@k is a widely used evaluation metric in recommender systems to measure the ranking quality of suggested items. NDCG@k takes into account both the relevance of recommendations and their positions in the ranked list, assigning higher importance to relevant items appearing earlier. To compute NDCG@k, first calculate the Discounted Cumulative Gain (DCG) up to position k, then normalize it by the ideal DCG (IDCG) obtained from the perfect ranking. Formally, for a list of recommendations $r_1, r_2, \ldots, r_k$, the DCG@k is defined as:

$$\text{DCG@k} = \sum_{i=1}^{k} \frac{2^{rel_i} - 1}{log_2(i+1)}, \tag{50}$$

where $rel_i$ is the relevance score of each item at position $i$. Then NDCG@k is the computed as:

$$\text{NDCG@k} = \frac{\text{DCG@k}}{\text{IDCG@k}}, \tag{51}$$

where IDCG@k is the maximum possible DCG@k for the ideal ordering of items.

Here's a polished version with the APT definition completed and implementation details clarified.

**Expected Free Discovery (EFD) [47]**  EFD quantifies how often a system surfaces relevant yet non-obvious items. We define

$$\text{EFD@}k = \frac{1}{k} \sum_{i=1}^{k} \text{disc}(i)\, p(\text{rel} \mid i, u)\, \log_2\big(p(i \mid \text{seen})\big), \tag{52}$$

where $\text{disc}(i)$ discounts lower-rank positions, $p(\text{rel} \mid i, u)$ is the user-specific relevance probability for the $i$-th recommended item, and $p(i \mid \text{seen})$ denotes the likelihood of encountering item $i$ given the user's previously seen items. In our implementation, we set $\text{disc}(i) = 0.85^{\,i-1}$; we estimate $p(\text{rel} \mid i, u)$ using the softmax over cosine similarities between item embeddings and the user embedding; and we approximate $p(i \mid \text{seen})$ via smoothed conditional co-occurrence counts between item $i$ and the user's interaction history (Laplace smoothing, $\alpha = 1$) computed on the training set.

**Average Percentage of Tail Items (APT) [2]**  APT measures the share of long-tail content in users' recommendation lists. Let $\mathcal{U}$ be the evaluation user set, $r_{u,i}$ the item at rank $i$ for user $u$, and $\mathcal{T}$ the tail set (defined below). We report APT as:

$$\text{APT@}k = \frac{1}{|\mathcal{U}|} \sum_{u \in \mathcal{U}} \left( \frac{1}{k} \sum_{i=1}^{k} \mathbf{1}[r_{u,i} \in \mathcal{T}] \right). \tag{53}$$

Tail items are determined by popularity on the training data: we compute item frequencies $\{f(j)\}$ and define $\mathcal{T} = \{j : f(j) \leq \tau\}$, where $\tau$ is a threshold chosen to mark the long tail (e.g., bottom 20% by frequency). We use this percentile-based thresholding and average uniformly over users.

## C   More Experimental Results

Here we present overall recommendation results on datasets Adressa, Yelp2018, ML10M and Globo extending results in Figure 5-8.

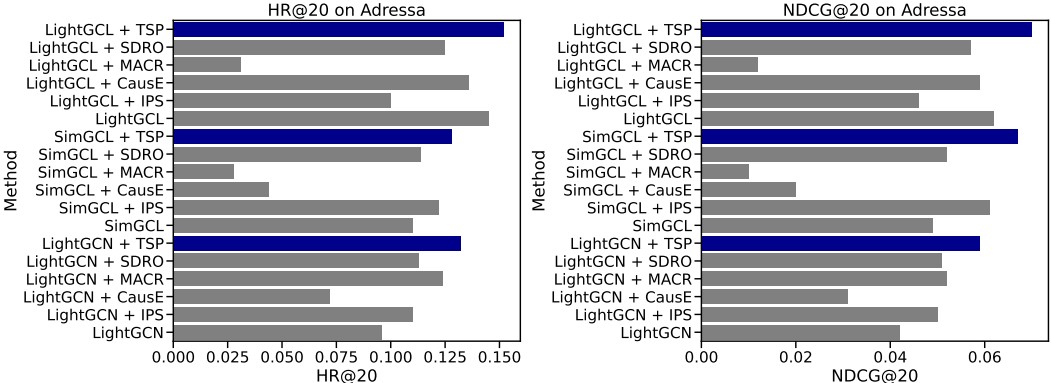

Figure 5: The overall performance of TSP and other baselines on Adressa dataset.

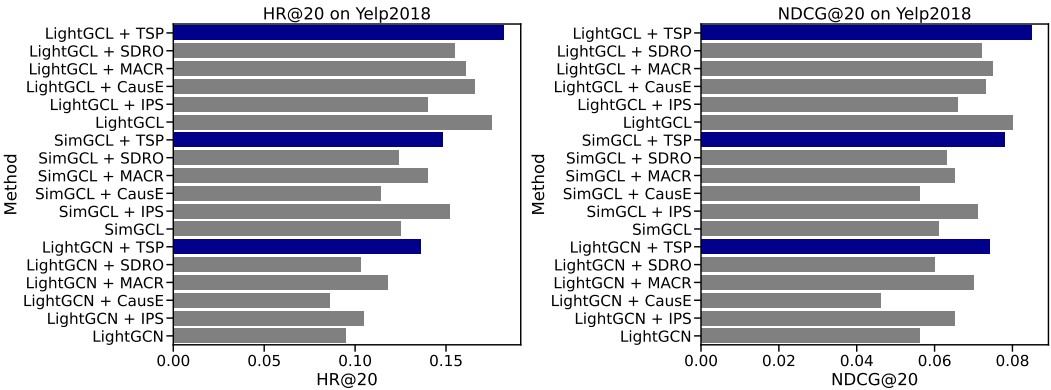

Figure 6: The overall performance of TSP and other baselines on Yelp2018 dataset.

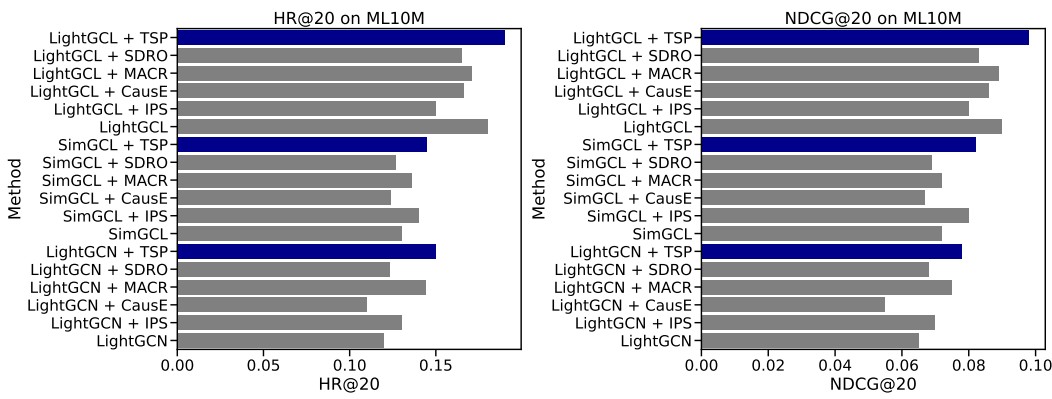

Figure 7: The overall performance of TSP and other baselines on ML10M dataset.

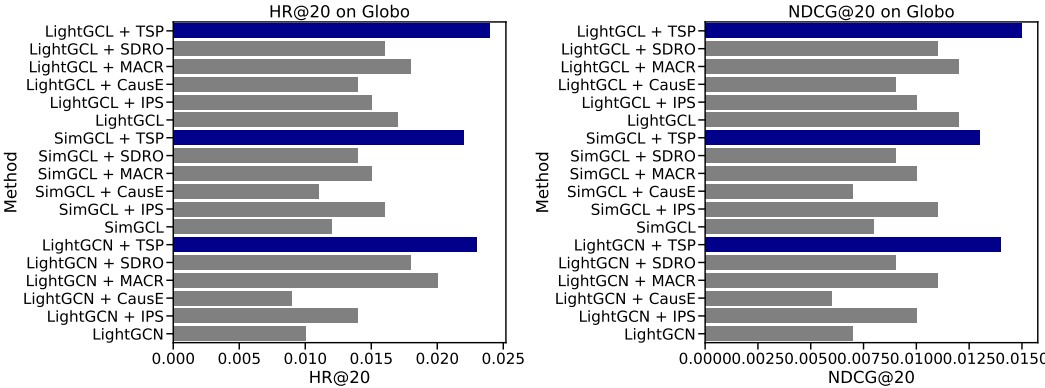

Figure 8: The overall performance of TSP and other baselines on Globo dataset.

# D  Examples of Simplicial Complexes

In a simplicial complex, the boundary of a $k$-simplex maps to its $(k-1)$-simplices (*i.e.,* the faces of the simplex). For a $k$-simplex $\sigma = \{v_0, v_1, \ldots, v_k\}$, the boundary $\partial\sigma$ is given by:

$$\partial\sigma = \sum_{i=0}^{k}(-1)^i\{v_0, v_1, \ldots, \hat{v}_i, \ldots, v_k\}, \tag{54}$$

where $\hat{v}_i$ indicates that vertex $v_i$ is omitted. This rule also determines the orientation of simplices in a SC boundary matrix as $-1$ or $1$, depending on the omitted $i$ of lower simplices.

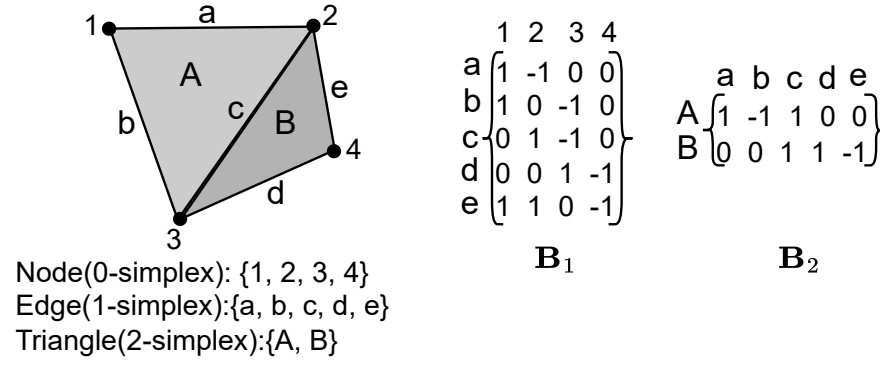

Figure 9: An example of simplices and boundary matrices.

We present an example simplicial complex consisting of 4 vertices in Figure 9. For triangle $A = \{1, 2, 3\}$, the edge $b = \{1, 3\}$ removes the $v_1$ in Equation 54, so its orientation (sign) in boundary matrix $\mathbf{B_2}$ is $-1$.

# E   TSP Algorithm

---

**Algorithm 1:** Test-time Simplicial Propagation

---

**Input:** Pretrained node embedding $\mathbf{X}_0$
Boundary matrices $\mathbf{B}_k$
Hodge Laplacians $\mathbf{L}_k$
Number of propagation layers $L$
Simplicial filter coefficient $\beta$
Maximum SC order $K$
**Output:** final node embeddings $\mathbf{X}_{rec}$

1 **for** $k = 1$ *to* $K$ **do**                                                                    /* Intra-Simplex Smoothing */

2     $\mathbf{X}_{\sigma^k}^{(0)} \leftarrow (\prod_{i=1}^{k} \mathbf{B}_i^T)\, \mathbf{X}_0$

3 **for** $k = 0$ *to* $K$ **do**                                                                    /* Inter-Simplex Propagation */

4     **for** $l = 1$ *to* $L$ **do**

5        $\mathbf{X}_{\sigma^k}^{(l+1)} \leftarrow (\mathbf{I} - \beta\mathbf{L}_k)\, \mathbf{X}_{\sigma^k}^{(l)}$

6 **for** $k = 1$ *to* $K$ **do**                                                                    /* Multi-order Fusion */

7     $\mathbf{X}_{0 \leftarrow k} \leftarrow (\prod_{i=1}^{k} \mathbf{B}_i)\, \mathbf{X}_{\sigma^k}^{(L)}$

8 $\mathbf{X}_{rec} \leftarrow \mathbf{X}_0 + \text{Mean}(\mathbf{X}_{0 \leftarrow 1}, \dots, \mathbf{X}_{0 \leftarrow K})$

9 **return** $\mathbf{X}_{rec}$

---

# F   Datasets

Five real-world recommendation datasets are used to evaluate the performance of TSP and other baselines with statistics in Table 5. The details of these datasets are as follows:

- **Adressa**[2] is a news dataset that includes news articles (in Norwegian) in connection with anonymized users.
- **Gowalla**[3] is a location-based social networking website where users share their locations by checking-in. The friendship network is undirected and was collected using their public API, and consists of 196,591 nodes and 950,327 edges.
- **Yelp2018**[4] is adopted from the 2018 edition of the yelp challenge, consisting of user interactions with local businesses like restaurants and bars. The 10-core setting (*i.e.,* only users and items with at least 10 interaction records are included) is adopted in order to ensure data quality.
- **Globo**[5] consists of users interactions logs (page views) from a news portal provided by Globo.com, the most popular news portal in Brazil. This dataset also includes rich user contextual attributes such as click_os, click_country, click_region and click_referrer_type.
- **ML10M**[6] is part of the movielens datasets with 10 million ratings by users. This dataset collects people's expressed preferences for movies as 0-5 star ratings. In experiments we follow the common practice of converting ratings to binary interactions.

# G   Implementation Details

Here we list the hyperparameter search intervals we use on evaluation sets across different datasets in Table 6. All experiments are conducted on a single Nvidia RTX A6000 Ada GPU with 48GB memory, with 3 repeated runs averaged for final results.

---

[2] https://reclab.idi.ntnu.no/dataset

[3] https://snap.stanford.edu/data/loc-gowalla.html

[4] https://www.yelp.com/dataset

[5] https://www.kaggle.com/gspmoreira/news-portal-user-interactions-by-globocom

[6] https://grouplens.org/datasets/movielens/10m

Table 5: Statistics of the datasets.

| Dataset | Users | Items | Interactions | Sparsity |
|---|---|---|---|---|
| Adressa | 13,485 | 744 | 116,321 | 0.011594 |
| Globo | 158,323 | 12,005 | 2,520,171 | 0.001326 |
| ML10M | 69,166 | 8,790 | 5,000,415 | 0.008225 |
| Yelp2018 | 31,668 | 38,048 | 1,561,406 | 0.001300 |
| Gowalla | 29,858 | 40,981 | 1,027,370 | 0.000840 |

Table 6: Search intervals of hyperparameters for TSP and other baselines.

| Models | Hyperparameters |
|---|---|
| LightGCN | reg weight $\alpha$: [1e-4, 5e-4, 1e-3, 5e-3, 1e-2] |
| LightGCL | CL loss weight $\lambda$: [1e-5,1e-6,1e-7,1e-8], rank of SVD $q$: [2, 5, 8, 10] |
| SimGCL | CL loss weight $\lambda$: [0.01, 0.05, 0.1, 0.2, 0.5, 1], noise magnitude $\epsilon$: [0, 0.01, 0.05, 0.1, 0.2, 0.5] |
| IPS | - |
| CausE | - |
| MACR | intermediate preference blocking $c$ :[10, 20, 30, 40, 50, 60], item loss weight $\alpha$ :[1e-5, 1e-4, 1e-3,1e-2], user loss weight $\beta$ :[1e-5, 1e-4, 1e-3,1e-2] |
| SDRO | - |
| TSP | maximum SC dimension $K$ :[3, 4, 5, 6], simplicial filter $\beta$ :[1e-3, 5e-3, 1e-2, 5e-2, 1e-1, 5e-1, 1], simplicial message passing layer $L$: [2, 3, 4] |

## H Ablation Study

We provide full performance of the variants in Section 5.5 across datasets with LightGCN backbone in Table 7.

Table 7: Ablation study across datasets. We report the overall and 20% tail item results of Top 20 performance. Best results are in **bold**, and second-best results are underlined.

| Scope | Variant | Adressa | | Yelp2018 | | Gowalla | | Globo | | ML10M | |
|---|---|---|---|---|---|---|---|---|---|---|---|
| | | R@20 | N@20 | R@20 | N@20 | R@20 | N@20 | R@20 | N@20 | R@20 | N@20 |
| Overall | TSP-Full | **0.132** | **0.059** | **0.020** | **0.018** | **0.082** | **0.058** | **0.027** | **0.019** | **0.027** | **0.018** |
| | TSP-SE | 0.108 | 0.046 | 0.010 | 0.006 | 0.056 | 0.041 | 0.016 | 0.012 | 0.015 | 0.011 |
| | LGN | 0.096 | 0.042 | 0.007 | 0.002 | 0.047 | 0.038 | 0.010 | 0.007 | 0.009 | 0.008 |
| Tail | TSP-Full | **0.055** | **0.024** | **0.009** | **0.008** | **0.038** | **0.021** | **0.024** | **0.013** | **0.013** | **0.008** |
| | TSP-SE | 0.024 | 0.013 | 0.006 | 0.003 | 0.017 | 0.009 | 0.007 | 0.006 | 0.006 | 0.003 |
| | LGN | 0.015 | 0.007 | 0.004 | 0.002 | 0.006 | 0.006 | 0.001 | 0.002 | 0.002 | 0.001 |

## I Lifting Graphs into Simplicial Complex

The implementation of lifting graphs into (clique) simplicial complexes centers on finding every clique in the graph, where a clique is a subset of vertices with every pair connected by an edge. Efficient algorithms such as Bron–Kerbosch are often employed for this task. Each detected clique of $k$ vertices is then interpreted as a simplex of dimension $k - 1$, so, for example, a triangle in the graph (a 3-clique) becomes a 2-dimensional simplex in the complex. To ensure the resulting structure is a simplicial complex, not only the maximal cliques but also all their subcliques should be included, since every subset of a clique is itself a clique and thus forms a lower-dimensional simplex.

We simplify this complicated process with TopoNetX [41], a suite specially designed to deal with topological data.

## J  Broader Impact and Limitations

In this paper we focus on addressing the topology bias in recommender systems, ensuring fairer recommendation results. This can directly contribute to the development of more equitable information access and distribution, promoting diversity and inclusivity in user experiences across various platforms. However, our approach is not without its limitations. Our current model solely concentrates on analyzing simplified graph message passing without incorporating weights or non-linear activations. While this simplification is often adopted to simplify the analysis on graph message passing, this may lead to incomplete understanding of how topology bias influences this process. We leave this for future work.

