# OpenReview forum: "How Does Topology Bias Distort Message Passing in Graph Recommender? A Dirichlet Energy Perspective"
_NeurIPS.cc/2025/Conference — NeurIPS 2025 poster_

### Official Review · Reviewer_QJ5t · 2025-06-23

**Clarity:** 3
**Significance:** 2
**Originality:** 2
**Rating:** 4
**Confidence:** 4

**Summary:**

Graph based recommender systems suffer from popularity bias. That is, the input graph (interaction graph between users and items) gets dominated by highly connected nodes which results in over-represenation of certain items. This results in reinforcing biases and fairness issues through the user-system feedback loop. This work analyses this topology bias through the lens of Dirichlet energy and show that message passing inherently minimises a global smoothing objective. Higher degree nodes accumulate larger embedding norms, a phenomenon called norm concentration. As a resolution, a test time simplicial propagation method is proposed which performs message passing over simplicial complexes with intra- and inter-simplex smoothing to avoid the dominance of certain nodes or substructures.

**Questions:**

See **weaknessess** above.

**Ethical Concerns:**

["NO or VERY MINOR ethics concerns only"]

**Final Justification:**

The authors have addressed my concerns and the experimental results strongly support their claims, I believe the current presentation does not adequately acknowledge existing work on over-smoothing and degree bias in the broader GNN literature. As I said in my review I believe the present work can be viewed as a special case of these established approaches applied to recommender systems. The theoretical contributions presented do not clearly articulate what is novel beyond the existing body of work on Dirichlet energy and over-smoothing. Given these concerns about theoretical positioning alongside the solid experimental validation, I have increased my score to 4.

**Limitations:**

Yes

**Quality:**

2

**Strengths And Weaknesses:**

**Strengths**

1. The Dirichlet energy perspective to analyse graph based recommender systems is interesting.

2. The formalization of norm concentration and how message passing favors highly connected nodes in recommender systems explaining over-representation of certain items is also fascinating.

3. The paper is well-written and easy to follow.


**Weaknessess**

**Major weaknesses**

1. I think the title of the paper should accurately reflect the contributions of the work. Currently, the title might prompt readers to think that the study is about topology bias in message passing GNNs covering more general form of GNNs such as GCNs, GATs, but the empirical analysis focuses specifically on **LightGCN-style architectures applied to graph based recommender systems**, which are simplified frameworks lacking feature transformations and non-linearities found in widely used GNNs. While I understand the creative liberty to have a catchy paper title, I think the title can be revised to better reflect the scope of the contributions.

2. The idea that message passing (MP) is inherently minimizing Dirichlet energy is not a new insight. Extensive work studying over-smoothing in GNNs (see some of the references listed below [1], [2], [3]) have already established this fact. The way the work is currently presented doesn't adequately acknowledge this foundation or clearly articulate what's novel beyond existing theoretical understanding. It would make sense to properly position this work’s contributions in light of these related works.

3. Although the norm concentration analysis is interesting, the idea that higher degree nodes receive larger message passing updates is obvious. For instance [4] also observes this phenomenon empirically (the current work also acknowledges this which is great). This leads again to my main issue which is properly discussing the contributions of the work in light of contemporary works. For instance works such as [9] and [10] treat this **degree bias/topology bias** more generally for GNNs and one can consider **norm concentration in LightGCN like architectures used specifically for recommender systems** as an application/special case of mitigating degree bias in GNNs ([4],[5],[6],[7],[8],[9],[10]).

4. The resolution proposed to mitigate this topology bias is test-time simplicial complex propagation approach which although seems to work based on presented experimental validation, it is usually expensive to compute. Currently, the work compares with only few debiasing methods but what about works such as [5],[6],[7],[8],[9] and [10] which might provide a cheaper way to mitigate **degree bias in the general framework of GNNs** and not restricted to recommender systems? It would be interesting to check if such proposed methods already address the identified issue of topology bias. And why a more computationally expensive approach of lifting graphs to simplices is a better/preferred approach?


**Minor weaknesses**

1. L26-27 maybe explain in 1-2 lines what exactly is Matthew effect?

2. The computational complexity for lifting the graphs into simplicial complex is missing.






**References**

1. A Note on Over-Smoothing for Graph Neural Networks. Cai et al, ICML 2020.

2. PairNorm: Tackling Oversmoothing in GNNs. Zhao et al, ICLR 2020

3. Not too little, not too much: a theoretical analysis of graph (over)smoothing. Keriven et al, 2022.

4. Test-Time Embedding Normalization for Popularity Bias Mitigation. Kim et al, CIKM 2023.

5. On Generalized Degree Fairness in Graph Neural Networks. Liu et al, AAAI 2023.

6. GRAPHPATCHER: Mitigating Degree Bias for Graph Neural Networks via Test-time Augmentation. Ju et al, NeurIPS 2023.

7. Tail-GNN: Tail-Node Graph Neural Networks. Liu et al, KDD, 2021.

8. Investigating and Mitigating Degree-Related Biases in Graph Convolutional Networks. Tang et al. CIKM 2020.

9. Theoretical and Empirical Insights into the Origins of Degree Bias in Graph Neural Networks. Subramonian et al, 2024.

10. Degree-based stratification of nodes in Graph Neural Networks. Ali et al, ACML 2023.

---

> ### Author Response · Authors · 2025-08-01
> **Author Rebuttal (1)**
>
> > **W1:** "I think the title of the paper should accurately reflect the contributions ..."
>
> **A1:** Thanks for raising this point. We agree that the scope of our paper is focused on graph-based recommender systems (RS) and this research focus could be  better reflected in the title. Since the LightGCN-style architecture is widely adopted by graph-based RS and state-of-the-art models, we focus primarily on presenting the bias issue presented in such models in empirical results. However, our theoretical framework and the induced method can be applied to general graph-based models. We are willing to modify our title to *"How Does Topology Bias Distort Message Passing in Graph Recommenders? A Dirichlet Energy Perspective"*.
>
> > **W2:** "The idea that message passing (MP) is inherently minimizing Dirichlet energy is not a new insight ..."
>
> **A2:** First, we want to clarify the theoretical differences of our paper with existing oversmoothing research papers. Dirichlet energy has long been used to study the oversmoothing problem in graph representation learning. while our paper shares the definition of aligning Dirichlet energy optimization with graph message passing, we present completely distinct theoretical analysis and methodology in our paper. Existing oversmoothing papers have established that the Dirichlet energy is exponentially minimized over the stacking of message passing layers and eventually result in the difference of node representations vanishing. On the contrary, our theoretical results show that the norm updates of node embeddings not only relate to their degrees (Corollary 3.1) but also are bounded by their Dirichlet energy (Lemma 3.2). Moreover, we extend the Dirchlet energy analysis to higher order simplicial complexes (SC) and present how message on SC can reduce the norm concentration phenomenon under such theoretical framework. We believe that our theoretical contributions are distinct from existing oversmoothing papers in both problem studied and the resulting methodology, showing originality and novelty in the study of topology bias. Nevertheless, we agree that clearly positioning our work in relation to existing studies on Dirichlet energy is important. We will add these details in the related work section.
>
> > **W3:** "Although the norm concentration analysis is interesting, the idea that higher degree nodes receive larger message ..."
>
> **A3:** We partially agree that the degree bias problem shares some similar aspects to our paper, but there are still some key differences we want to clarify. The degree bias often describes how nodes with high degrees in graph affect the classification results of other nodes, under the assumption of homophily, i.e., nodes closely connected tend to share the same class label.
> The debiasing process mainly focuses on improving the label accuracy for low degree nodes.
> However, we study how the topology bias in a graph distorts the message passing dynamics in graph-based recommenders, leading to popular items being dispropotionately recommended to users. The recommendation process can be regarded as a link prediction task in a heterogeneous user-item bipartite graph. We want our debiasing method to help niche items (low degree nodes) have an appropriate portion in final recommendation lists and thus provide results fit users' real interests. Moreover, the papers studying degree bias often deal with the node classification task in a semi-supervised manner, where labels are only available for a small subset of nodes. In contrast, nodes in graph-based RS are supervised by all their interactions, providing a supervision signal even for low-degree nodes. In summary, while they are both bias issues originated from unbalanced node degree distribution, they have distinct tasks, debiasing goals, bias formation mechanism and learning dynamics. Therefore, we believe our contributions are distinct and do not overlap with this line of research on degree bias
>
> > **W4:** "The resolution proposed to mitigate this topology bias is test-time simplicial complex ..."
>
> **A4:** Regarding the reviewer's concern about complexity, we want to further stress that our proposed TSP is designed as a *test-time* module that operates on a pre-trained model. The computational overhead of our TSP method, including the Hodge Laplacian calculation, does not affect the training efficiency of the underlying recommendation model. This one-time computation per test phase is a key advantage of our approach. As for the degree bias solutions, specially [1-6] mentioned by reviewer, as in our answers to Weakness 3, we want to kindly remind the reviewer that these methods all focus on node classification tasks. Due to distinct task, debiasing goals, learning dynamics and bias formation mechanism, we believe they are not directly applicable to topology bias in RS and thus can not serve as baselines for comparison with our proposed method.

---

> ### Author Response · Authors · 2025-08-01
> **Author Rebuttal (2)**
>
> > **W5:** "L26-27 maybe explain in 1-2 lines what exactly is Matthew effect?"
>
> **A5:** The Matthew effect describes the rich-get-richer phenomenon and is widely used in recommendation fairness works to represent the bias reinforcement loop. More specifically in topology bias, the popular items tend to have larger embeddings norms (Corollary 3.1) and updates (Lemma 3.2), which results in potentially higher prediction scores. This leads to more frequent recommendations, making these items even more popular. In this reinforcement loop, popular items can eventually dominate the recommendation lists and become disproportionately recommended to all users.
>
> **W6:** "The computational complexity for lifting the graphs into simplicial complex is missing."
>
>
> **A6:** The process of lifting a graph to a k-simplicial complex (e.g., finding all 3-cliques for 2-simplices) is a **one-time, offline preprocessing step** performed at test time. We apply the widely used clique mining Bron–Kerbosch [7]. For a graph with $n$ vertices, the worst-case complexity is $O(3^\wedge{(n/3)})$ [8, 9].
>
> **References**
>
>
> [1] On Generalized Degree Fairness in Graph Neural Networks. Liu et al, AAAI 2023.
>
> [2] GRAPHPATCHER: Mitigating Degree Bias for Graph Neural Networks via Test-time Augmentation. Ju et al, NeurIPS 2023.
>
> [3] Tail-GNN: Tail-Node Graph Neural Networks. Liu et al, KDD, 2021.
>
> [4] Investigating and Mitigating Degree-Related Biases in Graph Convolutional Networks. Tang et al. CIKM 2020.
>
> [5] Theoretical and Empirical Insights into the Origins of Degree Bias in Graph Neural Networks. Subramonian et al, 2024.
>
> [6] Degree-based stratification of nodes in Graph Neural Networks. Ali et al, ACML 2023.
>
> [7] Algorithm 457: finding all cliques of an undirected graph. Bron, C. et al, Communications of the ACM 1973.
>
> [8] On cliques in graphs. Moon J. W. et al, Israel Journal of 1965.
>
> [9] The worst-case time complexity for generating all maximal cliques and computational experiments. Tomita Etsuji et al, Theoretical Computer Science 2006.

---

> > ### Comment · Reviewer_QJ5t · 2025-08-01
> > **Thank you for the response.**
> >
> > Thank you for taking time to respond to my questions. Some clarifications about my concerns -
> >
> > 1. I still think the paper does not convincingly clarify the distinction between **degree-bias studied in regular GNNs** vs **topology-bias in LightGCN** like architectures studied in the context of recommender systems. The authors' rebuttal (A3) argues that the tasks (link prediction vs. node classification), debiasing goals, and supervision schemes are different. However, I think these distinctions are superficial since the underlying mechanism is identical: a message-passing scheme where update magnitude is directly influenced by node degree. The fact that this mechanism leads to over-recommending popular items in a bipartite graph (your "topology bias") is simply a different manifestation of the same fundamental issue that leads to poor performance on low-degree nodes in a classification task (the "degree bias").
> >
> > 2. Further, the method uses a pre-trained model and then constructs the semantic graph based on the learned representations and then proceeds to lift this semantic graph to simplicial complexes which is great that it works but kind of presents a circular problem - if we want to tackle topology bias in recommender graphs, do we not want to start at the beginning, for instance during training itself [1] or normalization techniques [4] or virtual nodes [2]? I think the paper fails to convince me - Why is it preferable to first learn biased representations and then apply a computationally expensive post-processing step (lifting to simplicial complexes), rather than addressing the bias at its source?

---

> > > ### Author Response · Authors · 2025-08-01
> > >
> > > Thank you for your engagement during the discussion period. We appreciate your time and thorough assessment, and your valuable feedback will help us improve the manuscript. We hope the following points address your concerns.
> > >
> > > > **Q1:** "I still think the paper does not convincingly clarify the distinction between ..."
> > >
> > > **A1:** We agree that the degree bias in node classification and the topology bias we study in recommender systems can be viewed collectively as the fundamental graph structure imbalance in different downstream graph tasks. We will include this similar line of work in the related work section to discuss the similarities and differences, which can indeed help to position our work. Additionally, we respectfully request that the reviewer gives greater consideration to the contributions presented in our manuscript. We would like to emphasize again that we are the first to **study graph and simplicial complexes message passing under unified Dirichlet energy framework** and **introduce message passing on simplicial complexes to mitigate bias in recommender systems** guided by our theoretical analysis.
> > >
> > > > **Q2:** "Further, the method uses a pre-trained model and then ..."
> > >
> > > **A2:**
> > > We understand that unbiased training is an intuitive solution. Indeed, most existing methods adopt this approach, typically incorporating debiasing strategies based on data distribution, user preference analysis, and similar considerations. However, we argue that training a model from scratch is often prohibitively expensive. In practical recommendation scenarios, the recommender system may already be trained on large-scale datasets, some encompassing billions of interactions. Therefore, debiasing methods that require full retraining are frequently impractical, and training-free test-time solutions are significantly more practical and appealing. This is the starting point of our work and a key distinction from existing approaches.
> > > Compared with training-based methods, our proposed TSP is indeed efficient. As shown in Table 3 of our paper, the one-time simplex lifting preprocessing for the ML10M dataset (with \~70k nodes and 5M edges) takes only 90 seconds. This is a small fraction of the 7-hour training time. Despite the worst-case complexity of simplex lifting, our TSP method achieves up to a 66% performance gain for tail nodes (Table 1) at less than 17% of the training time cost (Table 3), demonstrating its practical efficiency. In addition, another big advantage of our proposed method is that it is a plug-and-play module, demonstrating high flexibility and scalability. TSP can be integrated with various pre-trained graph recommenders without requiring further training or fine-tuning. We believe this model-agnostic nature makes our method widely applicable, allowing for efficient and flexible deployment across existing recommender systems.

---

> > > > ### Comment · Reviewer_QJ5t · 2025-08-01
> > > > **Questions answered**
> > > >
> > > > I would like to thank the authors for their prompt answers. This clarifies all of my doubts. I will be increasing my score. I would request the authors to kindly change the title to accurately reflect the paper's contributions and also discuss the related work on over-smoothing and discuss how fundamentally different/similar the current proposed approach is with respect to existing works as highlighted in my review.

---

> > > > > ### Author Response · Authors · 2025-08-01
> > > > >
> > > > > Thank you for your quick response and for your helpful suggestions. We are glad that we were able to answer your questions. We will modify the title and enhance the discussions of related work to better position our work.

---

### Official Review · Reviewer_CeVg · 2025-07-02

**Clarity:** 3
**Significance:** 3
**Originality:** 3
**Rating:** 5
**Confidence:** 4

**Summary:**

The paper provides, for the first time in the literature, a theoretical analysis of topology bias in message passing for graph-based recommender systems. Unlike similar previous analyses, mainly focusing on the node embeddings and gradients levels, the authors investigate this phenomenon under the lens of the Dirichlet energy.

To begin with, the paper theoretically demonstrates that performing the message passing is conceptually equivalent to setting a Dirichlet energy minimization problem. Secondly, they demonstrate how message passing leads to having a larger accumulation of embedding norms for high-degree nodes in the graph, and this phenomenon keeps amplifying the deeper we go in the message passing.

On such a basis, the authors propose a novel approach to tackle this issue, which interestingly runs in the inference step of the model's pipeline, thus it can be applied to any graph-based recommender systems during inference. The proposed pipeline creates a novel graph whose node connections are not necessarily based upon the topological structure of the original graph (which brings the bias information), but rather on the similarity of node embeddings conditioned on a hyperparameter $\theta$. After that, the framework transitions to a simplicial complex (intra-simplex smoothing module) and a propagation function to propagate embeddings across simplicials (inter-simplex propagation module). Finally, a fusion strategy is adopted to combine all node representations from different orders of simplices.

Extensive experimental results when applying the proposed solution during the inference process of three graph-based recommender models, against other debiasing baselines from the literature, largely demonstrate the efficacy of the proposed approach. This is true also across some recommendation datasets. Moreover, an ablation study on the main modules shows the goodness of the architectural choices made by the authors. Finally, an embedding distribution analysis empirically proves how the distribution is less concentrated in the case of the proposed solution, while a scalability study analyzes how the introduction of the methodology adds a very limited overhead on the simple inference of the original models.

**Questions:**

My main questions are related to the two weaknesses I outlined above.

Q1) Could the authors provide additional results when running the model RP3beta on the tested datasets?

Q2) How are the recommendation performance when considering additional standard bias and fairness metrics in the literature, such as the expected free discovery (EFD) and the average percentage of long-tail items (APLT)?

Q3) Do the authors have any intuition on how their theoretical analysis and subsequent methodology could be differently applied to the user side and the item side? In principle, the two sides are different and show different properties, also in the topological bias (i.e., low-high activeness on the platform for the user vs. popular/niche products for the items).

**Ethical Concerns:**

["NO or VERY MINOR ethics concerns only"]

**Final Justification:**

After the rebuttal phase, after reading the other reviews, and authors' responses to those, I decide to confirm my initial score I gave to the paper.

Specifically, all my concerns have been discussed and addressed by the authors.

Whatever the final outcome for this paper, I think it will benefit quite a lot from the fruitful discussion we had along with the authors, the reviewers, and the Area Chair. I believe the paper will improve its quality a lot in any case.

**Limitations:**

Yes.

**Paper Formatting Concerns:**

I do not see any major formatting issues with this paper.

**Quality:**

3

**Strengths And Weaknesses:**

**Strengths**

$\bullet$ The paper has the merit to not providing yet another graph-based recommendation model, but rather to analyze a critical issue in the current literature, topological bias, and disentangle it from a different theoretical perspective than other analyses provided in the recent literature.

$\bullet$ The paper is well-written and easy to follow.

$\bullet$ The methodology is quite clear and sound, as it is supported quite nicely by the background section. All theorems seem adequately sound and are rigorously proven in the Supplementary Material section.

$\bullet$ The proposed approach, which tackles the outlined issues, is nicely applied at inference time, thus not hurting the training of the models in any ways. The inference is also very limitedly hurt.

$\bullet$ Experimental settings are extensive with many evaluated dimensions. The code is released at review time, thus ensuring the complete reproducibility of the code and results presented in the paper.

**Weaknesses**

$\bullet$ The authors might have considered to include a further baseline (RP3Beta [\*]) which comes from a less-recent literature but works quite well on the user-item graph through random-walk and it is especially designed to not penalize less popular items. I think including this baseline as a standalone model to be compared against the other would provide a very nice complementary evaluation to the paper.

$\bullet$ In relation to the above, I think the authors might also consider to measure other recommendation measures which are usually adopted to assess the diversity in the recommendation lists [\*\*] and the presence of items from the long-tail [\*\*\*].

**References**

[\*] Christoffel et al., Blockbusters and Wallflowers: Accurate, Diverse, and Scalable Recommendations with Random Walks. RecSys 2015.

[\*\*] S. Vargas, P. Castells, Rank and relevance in novelty and diversity metrics for recommender systems, in: RecSys, ACM, 2011, pp. 109–116.

[\*\*\*] Himan Abdollahpouri, Robin Burke, and Bamshad Mobasher. 2017. Controlling Popularity Bias in Learning-to-Rank Recommendation. In RecSys. ACM, 42–46.

---

> ### Author Response · Authors · 2025-08-01
> **Author Rebuttal**
>
> > **W1:** "The authors might have considered to include a further baseline (RP3Beta ..."
> > **Q1:** "Could the authors provide additional results when running the model RP3beta ..."
>
> **A1:** We expand our experiments by adding the new baseline RP3beta compared with our method. The results of tail performance are listed below:
>
> | Models       | Adressa R@20 | Adressa N@20 | Gowalla R@20 | Gowalla N@20 | Yelp2018 R@20 | Yelp2018 N@20 | ML10M R@20 | ML10M N@20 | Globo R@20 | Globo N@20 |
> | ------------ | ------------ | ------------ | ------------ | ------------ | ------------- | ------------- | ---------- | ---------- | ---------- | ---------- |
> |       PR3beta       |    0.026          |0.009              |  0.013            |  0.009            |  0.006              |0.004                |   0.005         |    0.002        |      0.013      |0.007|
> | LightGCN+TSP |      0.055        |  0.024       |  0.038            |0.021|         0.009      |       0.008        |    0.013        |     0.008       |     0.024        | 0.013|
> | LightGCL+TSP |    0.075          |     **0.047**        |  0.050            |0.032              | **0.020**               | **0.019**               |   0.014         |0.010            |**0.032**            |       **0.018**     |
> | SimGCL+TSP             |**0.085**              |   0.046           |  **0.055**            |  **0.047**         |         0.015      |      0.014         |  **0.016**          |   **0.012**         |    0.025        |   0.017         |
>
> These additional result further present the strong performance of our proposed TSP method.
>
> > **W2:** "In relation to the above, I think the authors might also consider to measure other recommendation measures ..."
> > **Q2:** "How are the recommendation performance when considering additional standard bias and fairness metrics ..."
>
> **A2:** Thank you for your suggestions. We agree that the metrics mentioned can further present the debiasing ability of our proposed method. Due to limited rebuttal time we only evaluate methods on LightGCN backbone. We show these additional results  of EFD (with exponential discount $0.85^{k–1}$) and APLT of below:
>
> | Models         | Adressa EFD@20 | Adressa APT@20 | Gowalla EFD@20 | Gowalla APT@20 | Yelp2018 EFD@20 | Yelp2018 APT@20 | ML10M EFD@20 | ML10M APT@20 | Globo EFD@20 | Globo APT@20 |
> | -------------- | ----------- | ------------ | ----------- | ------------ | ------------ | ------------- | --------- | ---------- | --------- | ---------- |
> |PR3beta| 0.875 |0.125|0.840|0.142|0.804|0.095|0.741|0.072|0.755|0.068|
> | LightGCN       |   0.823          |       0.124       |   0.791          |         0.131     |         0.752     |    0.085           |      0.715     |          0.072  | 0.724          |    0.064        |
> | LightGCN+IPS   |     0.831        |     0.133         |    0.801         |0.156              |              0.764|     0.101          | 0.732           |0.080            | 0.714          |    0.073        |
> | LightGCN+CausE |  0.846           |     0.158         | 0.831            | 0.157             |  0.793            |0.105               |0.763           |      0.093      |                  0.735     |0.079|
> | LightGCN+MACR  | 0.901            |0.147| 0.856                          |           0.150   |   **0.814**           |     0.118          |   0.794        | 0.107|    0.756       |   0.121    |
> | LightGCN+SDRO  |    0.832         |0.136 |   0.781             |0.138  |  0.735 |       0.110        |      0.705     |    0.097        |  0.721         |       0.112     |
> | LightGCN+TSP   |       **0.923**      |**0.217**              |    **0.875**         |        **0.164**      |    0.813          |      **0.133**         |  **0.802**         |        **0.116**    |**0.773**           |  **0.149**         |
>
> The constant better performance of TSP in fairness metrics further prove its ability to mitigate the topology bias and produce balanced recommendation.
> > **Q3:** "Do the authors have any intuition on how their theoretical analysis and subsequent ..."
>
> **A3:** This is a very insightful question. In our current framework, we treat users and items symmetrically when constructing the semantic graph. However, the framework is flexible enough to handle them asymmetrically. For instance, one could use different similarity thresholds ($\theta$) for user-user and item-item pairs to reflect the different semantics and densities of these relationships. It would also be possible to construct simplices only on the item side to specifically focus on diversifying item representations. We believe this is a fascinating and promising direction for future research.

---

> > ### Comment · Reviewer_CeVg · 2025-08-01
> >
> > Dear Authors,
> >
> > Thanks for your time in answering to my raised weaknesses and questions.
> >
> > Your additional experiments with RP3Beta, as well as the diversity and popularity bias metrics, provide further empirical justifications to the goodness of your proposed approach.
> >
> > Moreover, I believe the intuitions you gave about how you could modify the formulation on the two sides (user and item) is quite nice and might represent an important avenue for future directions of your work.
> >
> > I do not have any other concerns regarding the paper. The answers you gave helped confirming the initial positive judgment I had on the paper. I will keep my score, which I believe is already quite high and mirrors the quality of your submission.

---

> > > ### Author Response · Authors · 2025-08-01
> > > **Thank you**
> > >
> > > Thanks for acknowledging our response and for your continued support!

---

### Official Review · Reviewer_HkJG · 2025-07-03

**Clarity:** 2
**Significance:** 2
**Originality:** 3
**Rating:** 4
**Confidence:** 3

**Summary:**

This paper studied graph-based recommender methods. The authors started by analyzing the common popularity bias issue in the graph-based methods, from a Dirichlet energy perspective. Combined with the theoretical analysis, the authors proposed TSP, a training framework for existing graph-based recommender methods to alleviate the effect of popularity bias. The proposed TSP contains multiple steps, such as the semantic graph reconstruction, which leverages a pretrained model to add in predicted edges, similar to graph structure learning, and a few following steps as detailed in Sec. 4.2.

**Questions:**

1. The term "semantic graph" is quite confusing. IIUC, the graph-based collaborative filtering methods doesn't use any semantic features, but instead learn ID embeddings for each nodes. The way TSP reconstructs the semantic graph is very similar to existing literature in graph structure learning and graph data augmentation, which were not discussed in this work. Hence I wonder why the reconstructed graph is named "semantic graph" when it's more of a "augmented graph"?

**Ethical Concerns:**

["NO or VERY MINOR ethics concerns only"]

**Final Justification:**

I appreciate the authors' rebuttal, which is very helpful with additional clarifications. So I raised my score.

**Limitations:**

see weakness

**Quality:**

3

**Strengths And Weaknesses:**

Strengths:
1. Popularity bias is a very important problem in collaborative filtering, which may even be enlarged in graph-based methods due to message passing. I'm glad to see papers trying to solve this critical issue.
2. This paper is well supported by theoretical analysis.

Weakness:
1. Graph-based (or message passing-based) recommender methods are often criticized by their limitations on efficiency when it comes to larger graphs [1]. While the authors has shown that the proposed TSP added limited overhead to lightGCN, considering that TSP has steps such as calculating the graph laplacian, I'm concerned on the efficiency of the proposed method.
2. Additional experiments on larger scale datasets would be appreciated.

[1] UltraGCN: Ultra Simplification of Graph Convolutional Networks for Recommendation, CIKM 2021

---

> ### Author Response · Authors · 2025-08-01
> **Author Rebuttal**
>
> >**W1:** "Graph-based (or message passing-based) recommender methods are ..."
>
>
> **A1:**  We agree that graph recommenders are often considered less efficient when trained on large graphs. However, our proposed TSP is designed as a *test-time* module that operates on a pre-trained model. The computational overhead of our TSP method, including the Hodge Laplacian calculation, does not affect the training efficiency of the underlying recommendation model. This one-time computation during test phase is a key advantage of our approach.
>
> Furthermore, the additional computational overhead introduced by our method is quite small. We have included a scalability analysis in Section 5.4 of our original paper, which demonstrates that our proposed TSP adds only a limited and acceptable overhead to the backbone models during inference time across datasets of varying sizes, as shown in Table 3.
>
>
> > **W2:** "Additional experiments on larger scale datasets would be appreciated."
>
> **A2:** We thank the reviewer for this suggestion and agree that scalability is a crucial aspect of graph-based models. While the definition of a "large-scale" dataset can vary, we have selected datasets that are considered substantial within the context of recent research. For example, the work cited by the reviewer [1] evaluates GCNs on datasets with up to around 100k nodes. Our experiments include datasets of comparable or greater scale, ranging from Gowalla (~14k nodes, ~100k interactions) to Globo (~160k nodes, ~2.5M interactions). We believe these results can already demonstrate our model's scalability across different dataset scales. Nevertheless, we are committed to providing a thorough empirical analysis. To best address the reviewer's concern, we would be grateful for any clarification on the specific scale or type of dataset that would be considered sufficient for a more comprehensive evaluation.
>
> > **Q:** "The term 'semantic graph' is quite confusing. IIUC, the graph-based ..."
>
> **A3:** We use the term "semantic graph" to emphasize that its structure is derived from the *semantic similarity* of node embeddings, which capture latent relationships (e.g., user preferences, item attributes) learned by the GNN, as opposed to the raw, explicit user-item interaction topology. This new topology is more reflective of the underlying semantic meaning of the nodes. We agree that this process can be considered as a form of graph augmentation or structure learning. However, our goal is not merely to augment the graph but to construct a new topology specifically guided by our Dirichlet energy analysis to mitigate topology bias.
>
> **References**
>
> [1] UltraGCN: Ultra Simplification of Graph Convolutional Networks for Recommendation, CIKM 2021

---

> > ### Comment · Reviewer_HkJG · 2025-08-08
> >
> > I appreciate the authors' rebuttal, which is very helpful with additional clarifications. Nevertheless, I don't think W2 and Q1 are well-addressed. For example, it doesn't really make sense to me to measure any "semantic similarity" between ID embeddings, which learns from collaborative signals.

---

> > > ### Author Response · Authors · 2025-08-09
> > >
> > > Thank you for your feedback.
> > >
> > > We agree that the name of the term "semantic graph" can cause certain confusion. In research fields like multi-modal recommendation, "semantic" typically refers to side information (e.g., text or images) rather than collaborative signals. This may lead to the assumption that our "semantic graph" should be built from such external data. In our work, we use this term to differentiate the latent relationships we want to uncover between nodes from the explicit connections in the existing interaction graph. Enhancing graph structures with learned embeddings or networks is a well-established practice in various domains [1-3]. Therefore, given our goal of a flexible, training-free solution, leveraging similar method by using the similarity of pre-trained embeddings as structural hints is an intuitive approach, which is also shown to be effective in our experiments. However, to avoid ambiguity, we will revise the manuscript and adopt the more precise term "latent graph" or "latent connections".
> > >
> > > Furthermore, to address your concerns and further validate our method, we have conducted additional experiments on the QK-video dataset from the Tenrec benchmark [4]. This is a large-scale, real-world dataset with 928,562 users, 1,189,341 items, and 37,823,609 interactions. With no specific statistics or dataset reference provided, we choose this industrial dataset which is more than 10 times larger than the largest dataset of our original experiments. The results of model performance and computational overhead are listed below. Due to the time constraints of the rebuttal period, we present results with only LightGCN as the backbone. We will include the full experimental results in the final version of our manuscript.
> > >
> > > | **Models**     | Tail Recall@20 | Tail NDCG@20 | overall Recall@20 | Overall NDCG@20 |
> > > | -------------- | -------------- | ------------ | ----------------- | --------------- |
> > > | LightGCN       |       0.001         |     0.002         |       0.063            |  0.049               |
> > > | LightGCN+IPS   |      0.008          |    0.011          |     0.069              |   0.057              |
> > > | LightGCN+CausE |       0.003         |      0.003        |     0.061              |0.050|
> > > | LightGCN+MACR  |     0.006           |     0.010         |    **0.081**           |     0.069            |
> > > | LightGCN+SDRO  |    0.004            |    0.006          |    0.070               |     0.053            |
> > > | LightGCN+TSP   |     **0.010**           |     **0.017**         |    **0.081**               |        **0.073**         |
> > >
> > >
> > >
> > > | LightGCN training time | TSP preprocessing time | LightGCN inference time | TSP inference time |
> > > | ---------------------- | ---------------------- | ----------------------- | ------------------ |
> > > |      39h                  |      4h                  |   0.853s                      |     0.881s               |
> > >
> > > We believe these additional results align with the conclusions in our original paper and further demonstrate the effectiveness and efficiency of our proposed TSP model on a large-scale, real-world dataset.
> > >
> > >
> > > **References**
> > >
> > > [1] Hierarchical Graph Convolutional Networks With Latent Structure Learning for Mechanical Fault Diagnosis. Kai Zhong et al, 2023.
> > >
> > > [2] Neural Common Neighbor with Completion for Link Prediction. Xiyuan Wang et al, ICLR 2024.
> > >
> > > [3] Locality-Aware Graph Rewiring in GNNs. Federico Barbero et al, ICLR 2024.
> > >
> > > [4] Tenrec: A Large-scale Multipurpose Benchmark Dataset for Recommender Systems. Guanghu Yuan et al, NeurIPS 2022.

---

> ### Comment · Area_Chair_W5yq · 2025-08-08
> **Engage with the rebuttal**
>
> Dear Reviewer HkJG,
> It would be useful if you could respond to the rebuttal of the authors to indicate if your concerns are satisfactorily answered.
>
> sincerely
> AC

---

### Note · Authors · 2025-08-14

Dear Reviewers and Area Chairs,

We thank all reviewers for the thorough review process.

We are delighted that reviewers all acknowledged the novelty and theoretical soundness of our work, describing our work as *"well supported by theoretical analysis"* (Reviewer HkJG), presenting a *"novel theoretical perspective"* with a *"clear and sound"* methodology (Reviewer CeVg), and offering an *"interesting Dirichlet energy perspective"* with a *"fascinating formalization"* of the topology bias problem (Reviewer QJ5t).

Detailed responses have been provided to each reviewer. We summarize the major revisions and clarifications here:

To address **Reviewer HkJG**'s concern about scalability, we conducted new experiments on the QK-video dataset from Tenrec benchmark, which is over 10x larger than any in our original submission. The additional results demonstrate the effectiveness and efficiency at scale of our method.

Following the suggestions of **Reviewer CeVg**, we have expanded our experimental results to include the PR3beta baseline and the EFD and APT metrics. We have also provided a discussion on the potential heterogeneous application of our framework to users and items.

In response to **Reviewer QJ5t**'s questions, we have provided discussions that explicitly differentiate our Dirichlet energy-based approach from related concepts like degree bias and oversmoothing. Furthermore, we clarified the practical motivation for our training-free, test-time design instead of re-training, better positioning our work's contribution.

Following our detailed rebuttal, we were glad that our responses successfully addressed the concerns of Reviewer CeVg and Reviewer QJ5t, leading to maintained positive score and increased rating. We believe our new large-scale experiments fully address the primary concern raised by Reviewer HkJG. Unfortunately, we have not heard back from Reviewer HkJG about the new experimental results during the discussion period.

We are fully committed to incorporating all suggested improvements into the final manuscript, including the new large-scale experiments, additional baseline and metrics, modifications of title and certain terms, and  the expanded discussions on related works.

Kind regards,

Authors of *How Does Topology Bias Distort Message Passing? A Dirichlet Energy Perspective*

---

### Decision · Program_Chairs · 2025-09-17

**Decision:**

Accept (poster)

**Comment:**

The paper addresses the problem of popularity bias—over-representation of popular items—in graph-based recommendation systems. It offers a fresh perspective by analyzing how graph topology influences message passing through the lens of Dirichlet energy. While the new ideas were appreciated, reviewers raised several concerns, particularly:
How the use of Dirichlet energy meaningfully differs from existing work on oversmoothing.
Evaluation of effectiveness on the long-tail of items.
Scalability of the proposed approach.
In response, the authors conducted additional experiments, which significantly strengthened the empirical section of the paper, leading to higher scores from some reviewers. However, during the discussion phase, the paper lacked a strong advocate. The ambivalence stemmed mainly from the perception that the theoretical contributions were modest. This impression was reinforced by the absence, in the main manuscript, of a clear summary of how the Dirichlet energy perspective connects to prior research on Graph Neural Networks. Although the rebuttal partially addressed this, questions about the extent of novelty remained.
Since a central motivation of the paper is to mitigate popularity bias, additional empirical evidence demonstrating improvements on long-tail recommendations would have been particularly impactful. Encouragingly, in the rebuttal, the authors presented results on one such dataset (as suggested by a reviewer), and the reviewer appreciated the results.

In summary, it will  be very difficult to make a strong recommendation for the paper. All the acceptance was contingent on the fact that substantive revision needs to be done. In case the paper is accepted the authors should mandatorily implement the changes discussed during rebuttal.